# Semantic-Enriched Latent Visual Reasoning

**Tianrun Xu** [1 2 3]  **Yue Sun** [4]  **Qixun Wang** [5]  **Jingyi Lu** [6]  **Yuan Wang** [3 7]  **Tianren Zhang** [1]  **Longteng Guo** [8 9]
**Fengyun Rao** [3]  **Jing LYU** [3]  **Feng Chen** [† 1]  **Jing Liu** [† 2 8 9]

## Abstract

Multimodal latent-space reasoning aims to replace explicit "thinking with images" by performing visual reasoning directly in a compact latent space. However, existing approaches largely rely on visual supervision and produce latent representations that lack sufficient semantic richness, limiting their ability to support diverse region-level reasoning tasks. In this work, we introduce **Semantic-Enriched Latent Visual Reasoning (SLVR)**, a two-stage learning framework that enriches latent representations with attribute-level visual semantics and aligns them with diverse reasoning objectives. In the first stage, SLVR learns semantically enriched region-centric latents under fine-grained attribute supervision. In the second stage, we design Multi-query Group Relative Policy Optimization (M-GRPO) to align latent representations across multiple queries grounded in the same region. To support this framework, we construct **SLV-Set**, comprising approximately 400K region-level attribute annotations and 800K multi-query question answering samples, and introduce **SV-QA**, a benchmark that evaluates latent reasoning under semantic variation. Experiments demonstrate that SLVR improves the robustness and semantic consistency of latent visual reasoning compared to existing baselines. Our code and datasets are available at https://github.com/tinnel123666888/slvr.

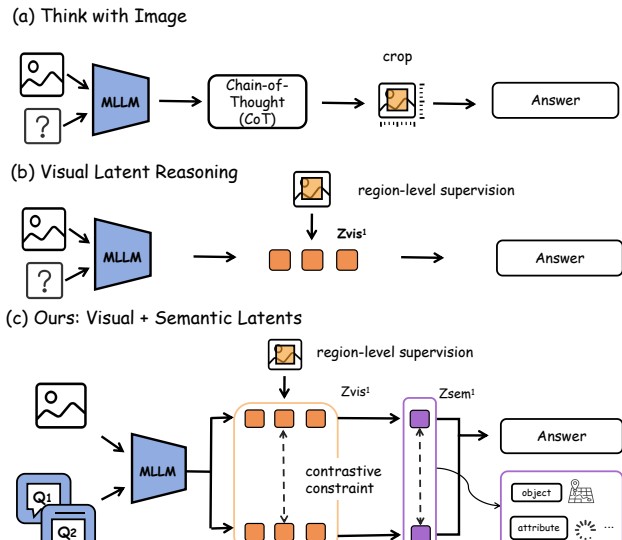

*Figure 1.* Conceptual comparison of (a) explicit reasoning or cropped evidence, (b) visual-only latent reasoning with visual supervision, and (c) our visually+semantically supervised latents with cross-question contrast.

## 1. Introduction

Vision-Language Models (VLMs) (Alayrac et al., 2022; Li et al., 2023; Zhu et al., 2023; Liu et al., 2023; 2024; Peng et al., 2023; Bai et al., 2023; Team et al., 2023; Chen et al., 2024; Xu et al., 2025a; Yan et al., 2026) have experienced significant progress in recent years, largely driven by the shift from direct inference approaches to more intermediate processes. Initially, VLMs were designed to directly output answers from visual inputs, a paradigm that relied on the straightforward interaction between images and questions. However, as the complexity of tasks increased, the limitations of this direct inference approach became evident. In response, recent developments have introduced a more sophisticated method, which first grounds the visual input to a common semantic space and then proceeds with reasoning. This "thinking with images" (Zheng et al., 2025; Fan et al., 2025; Xu et al., 2025b; Su et al., 2025)paradigm, while a significant step forward, still suffers from notable shortcomings. One of the key challenges is the reliance on explicit image cropping, which either focuses on cropped

[1]Department of Automation, Tsinghua University, Beijing, China [2]Zhongguancun Academy, Beijing, China [3]WeChat Vision, Tencent Inc, Beijing, China [4]China Agricultural University, Beijing, China [5]Peking University, Beijing, China [6]Beijing Institute of Technology, Beijing, China [7]Department of Electronic Engineering, Tsinghua University, Beijing, China [8]Institute of Automation, Chinese Academy of Sciences, Beijing, China [9]School of Artificial Intelligence, University of Chinese Academy of Sciences, Beijing, China. Correspondence to: Feng Chen <chenfeng@mail.tsinghua.edu.cn>, Jing Liu <jliu@nlpr.ia.ac.cn>.

*Proceedings of the 43rd International Conference on Machine Learning*, Seoul, South Korea. PMLR 306, 2026. Copyright 2026 by the author(s).

regions of interest or outputs target boxes purely in the text space, lacking deeper integration with the underlying visual information.

Recent work on Visual Latent Reasoning (Li et al., 2025a; Wang et al., 2025a) aims to replace explicit visual reasoning pipelines with compact latent representations. As shown in Fig. 1, existing approaches either rely on explicit chains of thought or cropped visual evidence (Fig. 1(a)), which are computationally inefficient, or compress visual information into a latent representation supervised purely by visual signals (Fig. 1(b)). While more efficient, visual supervision alone often leads latents to encode appearance-level cues without explicitly constraining what semantic content should be preserved for reasoning. As a result, existing approaches lack effective supervision that targets fine-grained and semantically rich latent representations, providing limited incentives for latents to capture detailed semantic structure. This highlights the need for a fine-grained attribute-level supervision paradigm, such as supervision over object properties and states, to support the learning of semantically enriched latent representations, as conceptually suggested in Fig. 1(c). Beyond semantic expressiveness, latent training is typically driven by a single task formulation, leaving no unified mechanism to align how the same enriched latent should support diverse query types and reasoning granularities. This calls for a multi-query optimization paradigm that aligns enriched latent representations with diverse reasoning objectives across granularities.

Based on the above motivations, we introduce **Semantic-Enriched Latent Visual Reasoning (SLVR)**, a two-stage framework that aims to enrich latent representations with fine-grained attribute semantics and align them with diverse reasoning demands. The central idea is to learn latent vectors that encode semantically rich attribute-level information of visual regions, such that they can flexibly support different tasks and reasoning granularities. In the first stage, SLVR introduces attribute-level semantic supervision to explicitly enrich latent representations. Latents are encouraged to capture fine-grained semantic attributes of visual regions, such as object properties and states, forming semantically expressive region-centric representations grounded in visual evidence. In the second stage, to effectively leverage these enriched latents across multiple tasks and reasoning granularities, we design Multi-query Group Relative Policy Optimization (M-GRPO). Instead of optimizing latents for a single query or task, M-GRPO jointly optimizes latent representations under multiple queries grounded in the same visual region, encouraging the model to consistently activate and utilize attribute-level semantics across diverse reasoning objectives while allowing task-specific variations.

To support the above framework, we construct two complementary datasets based on Visual-CoT (Shao et al., 2024).

The first is a region-level attribute dataset that provides standardized attribute descriptions for each key region, whose encoded text embeddings serve as fine-grained semantic supervision signals. The second is a multi-question dataset that associates each region with multiple questions probing different semantic aspects and reasoning granularities. To evaluate the effectiveness of visual-Semantic-Enriched latent representations under semantic variation, we further build multi-query visual reasoning benchmarks based on existing datasets, including $V^*$ (Wu & Xie, 2024) as well as HRBench-4K and HRBench-8K (Wang et al., 2025b), where each key region is queried from multiple semantic perspectives. Experiments on these benchmarks validate the effectiveness of aligning enriched latents with diverse reasoning objectives, leading to more robust and consistent latent visual reasoning.

Our main contributions are summarized as follows:

- We propose **SLVR** (Semantic-Enriched Visual Latent Reasoning), a two-stage learning framework that enriches latent representations with fine-grained attribute-level visual semantics and aligns them with diverse reasoning demands.

- To support the two-stage learning paradigm, we construct **SLV-Set**, which consists of two complementary components: approximately 400K region-level attribute annotations for semantic enrichment and 800K multi-query question answering samples for latent alignment.

- To evaluate visual-semantic latent reasoning under semantic variation, we introduce a new benchmark, **SV-QA**, which extends existing visual reasoning benchmarks by querying the same visual region from multiple semantic aspects and reasoning granularities.

- Extensive experiments on standard visual reasoning benchmarks and the proposed SV-QA demonstrate that SLVR consistently improves reasoning robustness and semantic consistency over existing latent reasoning baselines.

**Conflict of Interest Disclosure.** The authors declare no financial conflicts of interest related to this work.

## 2. Related work

### 2.1. Thinking with Images in Vision-Language Models

Recent work has explored explicit visual reasoning paradigms, commonly referred to as "thinking with images," where models perform reasoning through intermediate visual operations. COGCOM (Qi et al., 2024) formulates

visual reasoning as a sequence of explicit image manipulations, such as cropping and OCR, treating these operations as intermediate reasoning steps. GRIT (Fan et al., 2025) combines language reasoning with explicit region selection, using reinforcement learning to guide visual attention during inference. DeepEyes (Zheng et al., 2025) introduces adaptive zoom-in operations, allowing models to dynamically refine visual focus when evidence is uncertain. VisionReasoner (Liu et al., 2025b) unifies multiple perception tasks within a single framework by explicitly invoking visual operations during reasoning. MLLMs Know Where to Look (Zhang et al., 2025b) improves fine-grained perception through inference-time visual cropping strategies. Chain-of-Focus (Zhang et al., 2025c) learns multi-scale visual focus policies via reinforcement learning, enabling flexible zoom-in reasoning over cluttered scenes. V* (Wu & Xie, 2024) frames visual reasoning as guided visual search over high-resolution images. Visual-RFT (Liu et al., 2025c) refines visual grounding by reinforcement fine-tuning that aligns reasoning steps with visual evidence. In parallel, DeFacto (Xu et al., 2025b) leverages counterfactual visual reasoning to encourage models to attend to more faithful visual evidence. Despite their strong performance, these approaches rely on explicit chains of thought and function-like visual operations at inference time. Reasoning is carried out through repeated visual manipulations and tool invocations, which introduce significant computational overhead and limit efficiency. This motivates interest in latent-space reasoning paradigms that avoid explicit visual operations while retaining strong visual-semantic reasoning capabilities.

## 2.2. Latent Reasoning in Multimodal LLMs (MLLMs)

Recent advances in large language models have shifted the paradigm of reasoning from explicit, text-based chains toward implicit reasoning performed within latent spaces (Hao et al., 2025; Deng et al., 2026; Wang et al., 2025c; Chen et al., 2025; Wei et al., 2025).The paradigm of latent reasoning has also significantly advanced in Multimodal LLMs, moving from "text-only" reasoning to "visual" thinking (Zhang et al., 2025c; Zheng et al., 2025; Zhang et al., 2025b; Su et al., 2025). A critical contribution in this area is Latent Visual Reasoning (LVR) (Li et al., 2025a). LVR introduces a novel paradigm where the MLLM performs autoregressive reasoning directly within the visual embedding space. Alongside LVR, Latent Sketchpad (Zhang et al., 2025a) equips MLLMs with a "Vision Head" to interleave visual latent generation with text, effectively creating a "mental scratchpad" for spatial planning. Mirage (Yang et al., 2025) proposes "Machine Mental Imagery," using latent visual tokens optimized via reinforcement learning to simulate human-like mental visualization. DMLR (Liu et al., 2025a) introduces a "Reasoning Within the Mind" framework that dynamically injects visual features into the

reasoning chain based on confidence scores during test time. Additionally, Monet (Wang et al., 2025a) focuses on abstract visual thinking through a three-stage distillation pipeline and "Visual-Latent Policy Optimization" (VLPO) , while Mull-Tokens (Ray et al., 2025) proposes modality-agnostic thinking tokens to unify text and image reasoning. Overall, despite the rapid progress in latent-space reasoning for multimodal LLMs, the exploration of how to systematically enrich latent representations with rich, structured semantic information remains limited.

# 3. Dataset Construction for Semantic-Enriched Latent Reasoning

This section describes the construction of datasets. We build a unified data pipeline that provides fine-grained attribute-level semantic supervision, multi-query semantic variation, and standardized evaluation for latent visual reasoning. Specifically, we introduce SLV-Set, which is constructed based on the Visual-CoT (ViSCoT) (Shao et al., 2024) dataset and consists of two complementary components: an attribute-level semantic dataset with about 400K region descriptions and a multi-query dataset with roughly 800K question–answer pairs. In addition, we present SV-QA, a benchmark of 591 region-centric question–answer samples designed to evaluate latent reasoning robustness under semantic variation. The details are organized into three subsections: attribute-level semantic dataset construction (Sec. 3.1), multi-query dataset construction (Sec. 3.2), and SV-QA benchmark design (Sec. 3.3).

## 3.1. Attribute-Level Semantic Dataset

For attribute-level annotation, we employ Qwen3-VL-235B (Bai et al., 2025a) to construct a structured *region semantic profile* for each key visual region, following a region-centric prompting strategy illustrated in Fig. 2(a), resulting in approximately 400K region-level semantic profiles. The prompt is designed to produce standardized and interpretable attribute fields that summarize the local semantics of a region, including appearance attributes (e.g., entities, color, shape, material), action and interaction attributes, spatial attributes, and visible text content, forming a comprehensive semantic profile grounded in visual evidence. The resulting is then encoded into a single 4096-dimensional semantic embedding using a Qwen3 embedding model (Zhang et al., 2025d). Formally, each sample in the attribute-level dataset is represented as a quintuple $(q, i, b, \mathcal{A}, \mathbf{e})$, where $q$ denotes the associated question, $i$ the input image, $b$ the bounding box of the target region, $\mathcal{A}$ the set of region-level attribute descriptions, and $\mathbf{e} \in \mathbb{R}^{4096}$ the corresponding semantic embedding of the entire attribute set. Qualitative visualizations are provided in Appendix C. To assess the reliability of the automatically generated annotations, we

conducted a post-hoc human verification on the full SLV-Set. Annotators examined each sample by jointly inspecting the image, the region semantic profile, and the associated question–answer pairs, checking the consistency of grounding, semantic attributes, and QA alignment. In total, 29,457 erroneous annotations were identified, corresponding to an overall error rate of 7.29%. The detailed distribution of error types is summarized in Table 1. The most common issues are color mismatch (19.94%), action or pose error (14.55%), and spatial relation error (13.28%), followed by hallucinated or invisible attributes.

*Table 1.* Quantitative distribution of error types for SLVR across the SV-QA evaluation set.

| Error Category | Count | Ratio (%) |
| --- | --- | --- |
| Color mismatch | 5,873 | 19.94 |
| Action / pose error | 4,286 | 14.55 |
| Spatial relation error | 3,912 | 13.28 |
| Hallucinated attributes | 3,476 | 11.80 |
| Invisible objects | 3,158 | 10.72 |
| Incorrect held object | 2,941 | 9.98 |
| Ambiguous references | 2,834 | 9.62 |
| Overly vague | 2,267 | 7.70 |
| Others | 710 | 2.41 |
| **Total** | **29,457** | **100.00** |

### 3.2. Multi-Query Dataset

To construct the multi-query dataset, we use Qwen3-VL-235B to generate multiple questions grounded in the same visual region, following a structured prompting strategy described in Appendix B.2, resulting in approximately 800K region-centric question–answer pairs. As illustrated in Fig. 2(b), each target region serves as a semantic anchor from which two distinct questions are generated to probe different semantic aspects of the same visual evidence. Each sample is represented as $(i, q_1, q_2, b)$, where $i$ denotes the input image, $q_1$ and $q_2$ are two semantically distinct questions, and $b$ is the bounding box corresponding to the region most relevant to both questions. Qualitative visualizations of the constructed multi-query samples are provided in Appendix C.

### 3.3. SV-QA: A Benchmark for Semantic-Variation Reasoning

SV-QA is constructed to evaluate latent visual reasoning under semantic variation while keeping visual evidence fixed. We employ Qwen3-VL-235B to generate semantic-variation question pairs following the same prompting strategy described in Appendix B.4. Starting from existing visual reasoning benchmarks, including V* as well as HRBench-4K and HRBench-8K, we treat the original benchmark questions as $q_1$ and generate corresponding new questions $q_2$

that probe different semantic aspects of the same visual region. To ensure benchmark quality, all generated question pairs undergo manual inspection and refinement. During this process, 27 problematic cases were identified and revised, including 19 questions exhibiting excessive overlap with the original question and 8 questions containing hallucinated content unsupported by the visual region. The final benchmark contains 591 paired samples, consisting of 591 original questions and 591 manually verified new questions, each pair grounded in a shared target region but queried from distinct semantic perspectives. Formally, each SV-QA sample is represented as a tuple $(i, q_1, q_2, b)$, where $i$ denotes the image, $q_1$ and $q_2$ are semantically distinct questions, and $b$ is the bounding box of the shared target region. Qualitative visualizations of SV-QA samples are provided in Appendix D, enabling intuitive inspection of semantic variation under fixed visual evidence.

## 4. Semantic-Enriched Latent Visual Reasoning

This section introduces *Semantic-Enriched Latent Visual Reasoning (SLVR)*, a two-stage framework for learning region-centric latent representations that explicitly integrate fine-grained visual evidence with attribute-level semantic information. The goal of SLVR is to endow latent representations with structured semantics that can be consistently utilized across diverse reasoning queries grounded in the same visual region. Section 4.1 provides an overview of the framework and its core components. Section 4.2 presents the first stage, where structured visual and semantic latents are learned under supervised alignment. Section 4.3 introduces the second stage, which aligns these latents across multiple region-grounded queries via multi-query policy optimization.

### 4.1. Framework Overview

Fig. 3(a) illustrates the first stage of our framework, which focuses on learning structured latent representations. Given an input image and an original question, the model produces a region latent aligned with the corresponding visual evidence and a semantic latent encoding Semantic-Enriched region semantics. These latents are jointly supervised through region-level visual alignment, attribute prediction, and answer supervision, encouraging the latent representation to compactly encode both visual and semantic information.

In the second stage, shown in Fig. 3(b), we introduce Multi-query Group Relative Policy Optimization (M-GRPO) to refine the learned latent representations under multi-query constraints. Multiple questions referring to the same visual region are used to produce corresponding latent representations, which are optimized to remain consistent while preserving answer correctness. A stability regularization term further anchors the optimized latents to the distribution

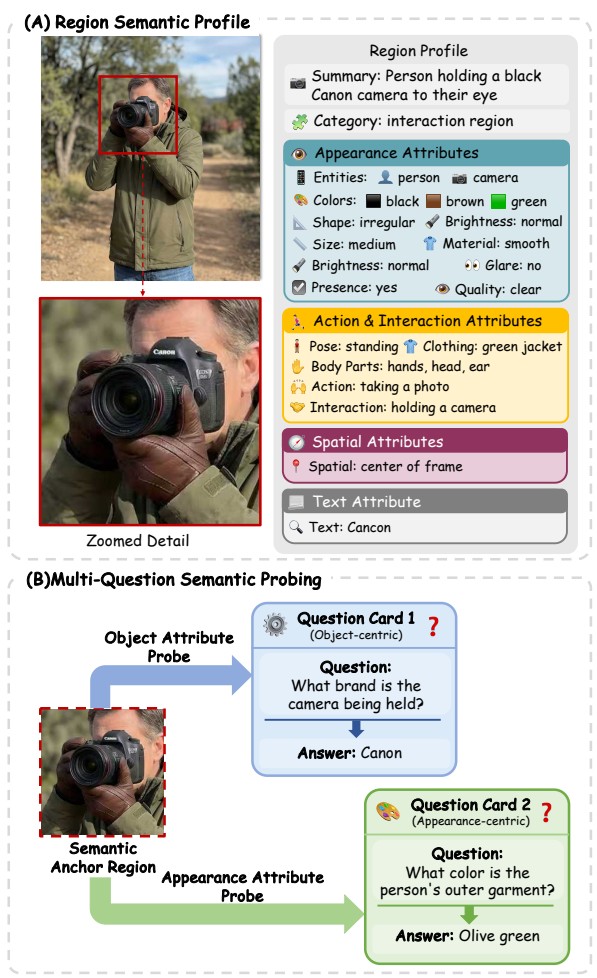

*Figure 2.* An illustration of our dataset construction.

learned in the first stage, preventing excessive drift during multi-query optimization.

## 4.2. Stage 1: Structured Latent Learning

The objective of the first stage is to learn region-centric latent representations that preserve fine-grained visual evidence while explicitly encoding attribute-level semantic information of the region. This is achieved by jointly supervising a region-level visual latent to retain local visual details and an additional semantic latent to capture structured region attributes, such as appearance, actions and interactions, and spatial properties.

**Inputs and Token Construction.** The model takes as input an image $I$ together with its original question $q$, which is associated with a key visual region. The image is encoded by a vision encoder into a sequence of visual patch features. A region of interest (ROI) is specified by an in-

dex set $\mathcal{I} = \{I_1, \ldots, I_{T_v}\}$, from which the corresponding encoder features $\{\mathbf{v}_t^{\mathrm{enc}}\}_{t=1}^{T_v}$ are selected. These ROI features are enclosed by the special tokens $\langle \mathtt{vis\_start} \rangle$ and $\langle \mathtt{vis\_end} \rangle$, forming a dedicated region-level latent segment whose hidden states constitute the region-level visual latent $\mathcal{H}_{\mathrm{vis}} = \{\mathbf{h}_t^{\mathrm{lat}}\}_{t=1}^{T_v}$. The textual tokens of the question are appended afterward for downstream reasoning and generation.

To enable explicit semantic supervision, we insert an additional special token $\langle \mathtt{sem} \rangle$ immediately after $\langle \mathtt{vis\_end} \rangle$. The hidden state corresponding to this token is denoted as a single semantic latent $\mathbf{z}_{\mathrm{sem}}$, which is designed to aggregate region-level semantic information.

**Region-Level Visual Latent and Alignment Objective.** Following the optimization principle of LVR, we directly align region-level latent hidden states with the visual features produced by the vision encoder.

Given a region of interest (ROI), the vision encoder yields a sequence of region-specific visual features $\{\mathbf{v}_t^{\mathrm{enc}}\}_{t=1}^{T_v}$. The hidden states corresponding to the ROI patch tokens form the region-level visual latent $\mathcal{H}_{\mathrm{vis}} = \{\mathbf{h}_t^{\mathrm{lat}}\}_{t=1}^{T_v}$. The visual alignment objective is defined as

$$\mathcal{H}_{\mathrm{vis}} = \{\mathbf{h}_t^{\mathrm{lat}}\}_{t=1}^{T_v}, \qquad \mathcal{L}_{\mathrm{vis}} = \sum_{t=1}^{T_v} \left\| \mathbf{h}_t^{\mathrm{lat}} - \mathbf{v}_t^{\mathrm{enc}} \right\|_2^2. \quad (1)$$

This direct feature-level supervision encourages the region-level latent tokens to faithfully preserve fine-grained visual evidence encoded by the vision encoder.

**Semantic Latent and Attribute Alignment.** The hidden state corresponding to the $\langle \mathtt{sem} \rangle$ token is taken as a semantic latent $z_{\mathrm{sem}}$. For each region, we use a single attribute-level semantic embedding $\mathbf{e}$ derived from the region semantic profile defined in Section 3.1 as explicit supervision, which encodes the aggregated attribute semantics associated with the target region.

$$\mathbf{e} \in \mathbb{R}^{4096}. \quad (2)$$

The semantic latent is projected into the same embedding space via a learnable projection head,

$$\hat{\mathbf{e}}_{\mathrm{sem}} = W z_{\mathrm{sem}}. \quad (3)$$

We align the projected semantic embedding with the attribute-level semantic supervision using a mean squared error loss,

$$\mathcal{L}_{\mathrm{sem}} = \mathrm{MSE}(\hat{\mathbf{e}}_{\mathrm{sem}}, \mathbf{e}) = \frac{1}{4096} \left\| \hat{\mathbf{e}}_{\mathrm{sem}} - \mathbf{e} \right\|_2^2. \quad (4)$$

**Overall Training Objective.** The overall training objective of the first stage combines the region-level visual reconstruction loss and the semantic alignment loss,

$$\mathcal{L}_{\mathrm{stage1}} = \mathcal{L}_{\mathrm{vis}} + \mathcal{L}_{\mathrm{sem}}, \quad (5)$$

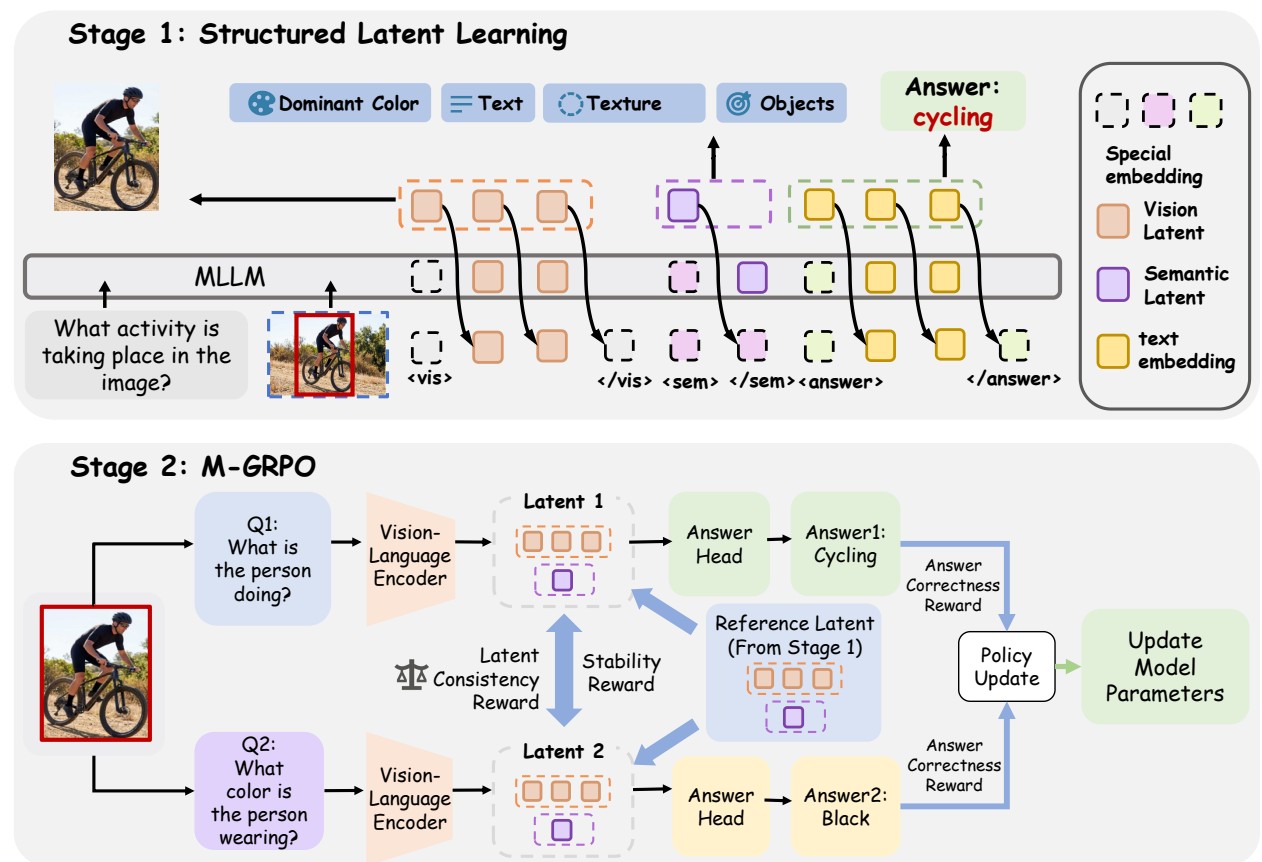

*Figure 3.* Overview of the proposed SLVR framework.

## 4.3. Stage 2: Multi-query Group Relative Policy Optimization.

The second stage aims to activate and align the semantically enriched latent representations learned in the first stage, so that they can be flexibly utilized to support diverse reasoning objectives and downstream tasks. Starting from latents that already encode rich attribute-level semantics, this stage introduces a multi-query optimization process that encourages consistent latent utilization under varying semantic demands while preserving their representational richness, thereby enabling reliable reasoning across different tasks, as illustrated in Fig. 3(b).

**Multi-query Group Relative Policy Optimization.** Multi-query Group Relative Policy Optimization (M-GRPO) is designed to encourage *both* the region-level visual latent and the semantic latent to encode comprehensive, Semantic-Enriched information of a visual region, rather than adapting narrowly to individual queries. As illustrated in Fig. 3(b), given multiple semantically distinct questions grounded in the same region, M-GRPO jointly refines the region latent and the semantic latent so that (i) each query can be answered correctly, (ii) the resulting latents remain consistent

across queries, and (iii) the optimized latents do not drift excessively from the Stage 1 reference distribution. Concretely, for each question $q_i$, the model produces a pair of latents

$$\left(H_{\text{vis}}^{(i)}, z_{\text{sem}}^{(i)}\right) = f(I, r, q_i), \qquad (6)$$

where $H_{\text{vis}}^{(i)} = \{h_t^{(i)}\}_{t=1}^{T_v}$ denotes the region-level visual latent segment and $z_{\text{sem}}^{(i)}$ denotes the semantic latent. M-GRPO performs policy updates over the model parameters that govern the generation of both latents under a composite reward consisting of three components.

*1. Answer Correctness Reward* encourages the latent pair to support correct reasoning for each corresponding query,

$$\mathcal{R}_{\text{ans}}^{(i)} = \mathbb{I}\left(\hat{y}_i = y_i\right), \qquad (7)$$

where $\hat{y}_i$ and $y_i$ denote the predicted and ground-truth answers for question $q_i$. In practice, answer correctness is determined by an external judge model (Qwen3-Max), which evaluates the model output against the reference answer and assigns a binary reward.

*2. Latent Consistency Reward* enforces cross-query consistency for *both* the region-level visual latent and the semantic

latent. We define the pairwise consistency reward as

$$\mathcal{R}_{\text{cons}} = -\sum_{i \neq j} \left( \lambda_{\text{sem}} \left\| z_{\text{sem}}^{(i)} - z_{\text{sem}}^{(j)} \right\|_2 \right.$$
$$\left. + \lambda_{\text{vis}} \frac{1}{T_v} \sum_{t=1}^{T_v} \left\| h_t^{(i)} - h_t^{(j)} \right\|_2 \right). \quad (8)$$

where $\lambda_{\text{sem}}$ and $\lambda_{\text{vis}}$ balance semantic-level and region-level visual consistency, respectively.

*3. Stability Regularization.* To prevent excessive drift during multi-query optimization, we anchor *both* latents to their Stage 1 references using a margin-based formulation. Concretely, after Stage 1 training converges, we freeze the Stage 1 model and run a forward pass on the corresponding samples to extract their semantic and region-level latents, which are then stored as reference anchors for Stage 2 optimization. Let $\bar{z}_{\text{sem}}$ and $\bar{H}_{\text{vis}} = \{\bar{h}_t\}_{t=1}^{T_v}$ denote the reference semantic and region latents obtained from the first-stage model. We define

$$\mathcal{R}_{\text{stab}}^{(i)} = -\max\left( 0, \left\| z_{\text{sem}}^{(i)} - \bar{z}_{\text{sem}} \right\|_2 - \tau_{\text{sem}} \right)$$
$$-\max\left( 0, \frac{1}{T_v} \sum_{t=1}^{T_v} \left\| h_t^{(i)} - \bar{h}_t \right\|_2 - \tau_{\text{vis}} \right). \quad (9)$$

The stored Stage 1 latents act as stable targets that constrain Stage 2 updates to remain within a bounded neighborhood, while still allowing necessary adaptation under multi-query supervision. The tolerance margins $\tau_{\text{sem}}$ and $\tau_{\text{vis}}$ control the allowable deviation range for the semantic latent and the region-level visual latent, respectively.

**Optimization Objective.** M-GRPO adopts a Group Relative Policy Optimization (GRPO) (Guo et al., 2025) objective to update the latent-generation policy over groups of region-grounded queries. For each query group $\mathcal{Q}(r)$, the optimization objective is defined as

$$\mathcal{J}_{\text{M-GRPO}}(\theta) = \mathbb{E}_{\substack{(I,r) \sim \mathcal{D}, (q_1,q_2) = \mathcal{Q}(I,r), \\ \{o_{i,g,t}\} \underset{i=1,2}{\sim} \pi_{\theta_{\text{old}}}(\cdot|I,r,q_i) \\ g=1,\dots,G \\ t=1,\dots,|\mathcal{O}_{i,g}|}}$$

$$\frac{1}{2} \sum_{i=1}^{2} \frac{1}{G} \sum_{g=1}^{G} \frac{1}{|\mathcal{O}_{i,g}|} \sum_{t=1}^{|\mathcal{O}_{i,g}|} \min\left( \rho_{i,g,t}(\theta) \, \hat{A}_{i,g,t}, \right.$$

$$\left. \text{clip}\left( \rho_{i,g,t}(\theta), 1-\epsilon, 1+\epsilon \right) \hat{A}_{i,g,t} \right)$$

$$- \beta \frac{1}{2} \sum_{i=1}^{2} D_{\text{KL}}\Big( \pi_\theta(\cdot \mid I, r, q_i) \, \|$$

$$\pi_{\theta_{\text{old}}}(\cdot \mid I, r, q_i) \Big) \quad (10)$$

Here each group is anchored by a fixed region $(I, r)$ and contains exactly two pre-generated queries $(q_1, q_2)$. For each query $q_i$, we sample $G$ rollouts, and the $g$-th rollout is a token sequence $\mathcal{O}_{i,g} = \{o_{i,g,t}\}_{t=1}^{O}$. In practice, the two queries in each group share the same underlying region and latent initialization, and are optimized jointly within a single GRPO update step. This grouped formulation encourages the latent-generation policy to produce representations that are simultaneously effective for multiple semantic queries over the same region, rather than optimizing each query independently. Following GRPO, we define the importance sampling ratio at token level as

$$\rho_{i,g,t}(\theta) = \frac{\pi_\theta\left( o_{i,g,t} \mid q_i, I, r, H_{\text{vis}}^{(i,g)}, z_{\text{sem}}^{(i,g)}, o_{i,g,<t} \right)}{\pi_{\theta_{\text{old}}}\left( o_{i,g,t} \mid q_i, I, r, H_{\text{vis}}^{(i,g)}, z_{\text{sem}}^{(i,g)}, o_{i,g,<t} \right)}$$
$$\quad (11)$$

## 5. Experiments

### 5.1. Experimental Setup

**Evaluation Benchmarks.** We evaluate our method on a diverse set of multimodal reasoning benchmarks covering general, document-centric, and text-intensive scenarios, including OKVQA (Marino et al., 2019), GQA (Hudson & Manning, 2019), VizWiz (Gurari et al., 2018), ChartQA (Masry et al., 2022), TextVQA (Singh et al., 2019), and AI2D (Kembhavi et al., 2016). In addition, we introduce SV-QA, a curated benchmark constructed as described in Section 3.3 to assess semantic-enriched visual reasoning under high-resolution and fine-grained settings. Qualitative visualizations of SV-QA are provided in Appendix D.

**Evaluation Baselines.** We compare our method with representative baselines that can be broadly grouped into two categories. The first category follows a think-with-image paradigm, where visual information is repeatedly accessed during reasoning, including GRIT (Fan et al., 2025), DEEP-EYES (Zheng et al., 2025), and VISUAL-SR1 (Li et al., 2025c). The second category focuses on latent reasoning, where intermediate representations are used to support inference, including Latent Visual (Li et al., 2025a) Reasoning (LVR) and Monet (Wang et al., 2025a).

### 5.2. Main Results

Table 2 summarizes the evaluation results on standard VQA benchmarks. Overall, SLVR demonstrates strong and consistent performance across diverse datasets, outperforming prior latent-reasoning approaches and remaining competitive with think-with-image baselines, especially on OKVQA, VizWiz, and ChartQA.

Table 3 further examines region-centric reasoning on the SV-QA benchmark. Compared with Qwen2.5-VL and the

*Table 2.* Evaluation results on VQA benchmarks (accuracy, %).

| Model | Reasoning Paradigm | OKVQA | GQA | VizWiz | ChartQA | TextVQA | AI2D |
|---|---|---|---|---|---|---|---|
| Qwen2.5-VL-7B (Bai et al., 2025b) | Text-only | 58.9 | 53.2 | 54.1 | 74.4 | 79.1 | 69.5 |
| GRIT (Fan et al., 2025) | Text CoT | 43.1 | 54.8 | 39.4 | 24.6 | 60.8 | 75.5 |
| DeepEyes (Zheng et al., 2025) | Text + Visual Cropping | 40.0 | 45.5 | 32.2 | – | 40.4 | 37.0 |
| Visual-SR1 (Li et al., 2025c) | Text CoT | 49.0 | 52.3 | 35.8 | 71.7 | 70.1 | 69.0 |
| LVR (Li et al., 2025a) | Visual Latent | 50.6 | **57.4** | 33.1 | 64.4 | 75.1 | **77.3** |
| **SLVR-7B** | Visual + Semantic Latent | **61.8** | 55.6 | **57.8** | **77.2** | **79.3** | 76.0 |
| *Gain over LVR* | | **+11.2** | -1.8 | **+24.7** | **+12.8** | **+4.2** | -1.3 |

*Table 3.* Single-question accuracy under different latent settings on our **SV-QA** dataset. Q1 and Q2 denote two region-centric questions; **Both Correct** indicates the percentage of samples where both answers are correct.

| Setting | Reasoning Paradigm | V* | | | HRBench-4K | | | HRBench-8K | | |
|---|---|---|---|---|---|---|---|---|---|---|
| | | Q1 | Q2 | **Both** | Q1 | Q2 | **Both** | Q1 | Q2 | **Both** |
| Qwen2.5-VL (Bai et al., 2025b) | Text-only | 76.4 | 55.5 | 44.0 | 60.5 | 74.1 | 45.8 | 53.5 | 66.1 | 37.5 |
| GRIT (Fan et al., 2025) | Text CoT | 69.6 | 61.8 | 49.7 | 58.4 | 70.4 | 48.6 | 49.1 | 61.9 | 38.6 |
| DeepEyes (Zheng et al., 2025) | Text + Visual Cropping | **83.3** | 79.1 | **70.2** | **71.3** | 79.3 | 60.6 | **65.1** | 72.4 | 49.0 |
| Visual-SR1 (Li et al., 2025c) | Text CoT | 75.9 | 73.3 | 62.8 | 63.9 | 76.4 | 54.6 | 56.3 | 67.4 | 44.4 |
| LVR (Li et al., 2025a) | Visual Latent | 81.7 | 77.5 | 65.4 | 70.1 | 80.4 | 57.9 | 62.9 | 71.1 | 46.6 |
| **SLVR-7B** | Visual + Semantic Latent | 82.2 | **80.1** | 69.1 | 70.4 | **82.8** | **61.1** | 62.5 | **73.6** | **50.6** |
| *Gain over LVR* | | +0.5 | +2.6 | +3.7 | +0.3 | +2.4 | +3.2 | −0.4 | +2.5 | +4.0 |

original latent reasoning baseline (LVR (Li et al., 2025a)), SLVR achieves consistently better performance across multiple semantic dimensions, with improvements observed on both single-question accuracy and the Both Correct metric over V*, HRBench-4K, and HRBench-8K.

In addition, we provide a more comprehensive comparison against representative state-of-the-art models across a broader set of multimodal benchmarks in Appendix A.

**Comparison on Public VQA Benchmarks.** Table 2 compares our method with think-with-image approaches and latent visual reasoning methods on public VQA benchmarks. As shown in Table 2, SLVR achieves competitive or superior performance across most benchmarks, outperforming prior latent reasoning methods and matching or exceeding think-with-image baselines on datasets such as OKVQA, VizWiz, and ChartQA. To further assess whether the learned semantic-enriched latents generalize beyond standard VQA settings, we evaluate SLVR on VisualPuzzles (Song et al., 2025). As shown in Table 5, SLVR consistently outperforms both Qwen2.5-VL and the LVR baseline, achieving the best overall accuracy (34.2 vs. 29.4 for LVR, +4.8). The improvements are particularly pronounced on Deductive (+5.0 over LVR) and Spatial (+7.4 over LVR) categories, suggesting that the attribute-level semantic supervision helps the model capture structured relational and spatial cues that purely visual-supervised latents tend to miss.

**Results on SV-QA.** Table 3 reports results on the SV-QA benchmark, where each visual region is queried from two distinct semantic perspectives. SLVR consistently outperforms Qwen2.5-VL and improves joint correctness over LVR across V*, HRBench-4K, and HRBench-8K, while maintaining comparable or better single-question accuracy. Although DeepEyes achieves higher Q1 accuracy on V* and HRBench, this advantage primarily reflects its reliance on explicit visual cropping during inference, which incurs substantially higher computational cost. In contrast, SLVR achieves competitive Q1 performance and the best Q2 and Both Correct results on HRBench-4K and HRBench-8K purely through latent-space reasoning, without any explicit visual operations. The gains on the Both Correct metric, which jointly evaluates two semantically distinct questions over the same region, indicate that the learned latents encode more stable and semantically richer information that can be consistently utilized across multiple reasoning perspectives.

### 5.3. Ablation Studies

We conduct ablation studies from two complementary perspectives. First, we compare two reasoning paradigms—text-only chain-of-thought and latent reasoning—under matched training data, allowing us to isolate the inherent advantages of latent-space reasoning from those of our specific design (Table 4, upper block). Second, we analyze the contribution of different training components within the latent reasoning family using an additive design (Table 4, lower block). Starting from the LVR baseline,

*Table 4.* Additive ablation results on the **SV-QA** dataset. **Multi-Q** denotes training with multiple questions per instance (more training signals), while **M-GRPO** further introduces explicit latent-level consistency constraints beyond standard GRPO. Q1 and Q2 denote two region-centric questions; **Both** indicates the percentage of samples where both answers are correct.

| Setting | Components | | | | | V* | | | HRBench-4K | | | HRBench-8K | | |
|---|---|---|---|---|---|---|---|---|---|---|---|---|---|---|
| | LVR | Stage 1 | Multi-Q | GRPO | M-GRPO | Q1 | Q2 | Both | Q1 | Q2 | Both | Q1 | Q2 | Both |
| *Text-only Baselines (Qwen2.5-VL-7B)* | | | | | | | | | | | | | | |
| SFT (single-Q) | – | ✓ | – | – | – | 77.5 | 66.5 | 51.8 | 67.9 | 80.1 | 52.8 | 61.2 | 72.9 | 43.5 |
| SFT (multi-Q) | – | ✓ | ✓ | – | – | 79.6 | 69.6 | 57.1 | 66.5 | 78.4 | 50.1 | 60.5 | 71.1 | 41.1 |
| SFT + GRPO (multi-Q) | – | ✓ | ✓ | ✓ | – | 80.6 | 75.4 | 58.6 | **70.5** | 82.5 | 55.5 | **64.9** | **76.5** | 47.8 |
| *Latent Reasoning Models (LVR & SLVR)* | | | | | | | | | | | | | | |
| LVR | ✓ | – | – | – | – | 81.7 | 77.5 | 65.4 | 70.1 | 80.4 | 57.9 | 62.9 | 71.1 | 46.6 |
| + Stage 1 | ✓ | ✓ | – | – | – | **82.2** | 75.4 | 67.5 | 67.4 | 77.9 | 59.6 | 60.8 | 72.1 | 46.9 |
| + GRPO (Single-Q) | ✓ | ✓ | – | ✓ | – | 78.5 | 78.5 | 62.8 | 69.9 | 82.1 | 57.6 | 62.6 | 70.5 | 49.3 |
| + Multi-Q | ✓ | ✓ | ✓ | ✓ | – | 81.7 | 79.6 | **68.1** | **70.5** | 82.0 | 60.3 | 62.4 | 72.5 | 49.9 |
| LVR + Multi-Q | ✓ | – | ✓ | ✓ | – | 81.7 | 79.1 | 67.5 | 70.3 | 81.4 | 60.0 | 63.1 | 71.8 | 49.4 |
| **SLVR-7B (Full)** | ✓ | ✓ | ✓ | ✓ | ✓ | **82.2** | **80.1** | **69.1** | 70.4 | **82.8** | **61.1** | 62.5 | 73.6 | **50.6** |

*Table 5.* Performance comparison on VisualPuzzles across five reasoning categories (accuracy, %).

| Model | Algorithmic | Analogical | Deductive | Inductive | Spatial | Overall |
|---|---|---|---|---|---|---|
| Qwen2.5-VL | 35.9 | 26.1 | 35.5 | **28.7** | 21.3 | 29.2 |
| LVR | 31.3 | 25.6 | 40.5 | 24.4 | 26.2 | 29.4 |
| **SLVR (Ours)** | **37.4** | **28.0** | **45.5** | 26.3 | **33.6** | **34.2** |

we progressively introduce (i) semantic latent supervision via Stage 1 training ($z_{sem}$ + SFT), (ii) single-question policy optimization with standard GRPO, (iii) multi-question training that increases supervision signals without explicit grouping constraints, and (iv) the full SLVR model with M-GRPO, which enforces latent-level consistency across paired questions.

**Text-only vs. Latent Reasoning Baselines.** The upper block of Table 4 reports three text-only baselines built on Qwen2.5-VL-7B that share the same training data as our latent variants but reason purely in the text space. Starting from standard supervised fine-tuning on single-question data (SFT (single-Q)), adding our multi-question data (SFT (multi-Q)) brings noticeable gains on V*, indicating that exposing the model to multiple semantic perspectives over the same region provides richer supervision signals. Further introducing reinforcement learning on the combined data (SFT + GRPO (multi-Q)) yields the strongest text-only results, particularly on HRBench-4K Q1 (70.5) and HRBench-8K Q1 (64.9). However, even the best text-only configuration consistently lags behind the latent reasoning variants on the *Both Correct* metric across all three datasets, suggesting that explicit textual chain-of-thought struggles to maintain consistent grounding when the same region is queried from different semantic angles.

**Takeaway: Effects of Training Components.** The lower block of Table 4 summarizes the effects of progressively introducing different training components on the SV-QA benchmark. Starting from the LVR baseline, adding Stage 1 semantic latent supervision improves joint correctness, while single-question GRPO brings limited or mixed gains. Training with multiple questions further enhances performance, indicating the benefit of richer supervision signals. The full SLVR model with M-GRPO consistently achieves the strongest and most balanced results, highlighting the importance of explicitly enforcing latent-level consistency.

## Impact Statement

This paper presents work whose goal is to advance the field of machine learning. There are many potential societal consequences of our work, none of which we feel must be specifically highlighted here.

## Acknowledgements

This work was supported by the Zhongguancun Academy (Grant No. C20250505), the National Key Research and Development Program of China (No. 2024YDLN0006), the National Key Research and Development Program of China under STI 2030—Major Projects (No. 2021ZD0200300), and the National Natural Science Foundation of China (Nos. 62531026, 62437001).

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

# A. Additional Comparison on SoTA Benchmarks

We further compare our method with representative state-of-the-art models on a diverse set of multimodal benchmarks, as summarized in Table 6. The results show that our approach consistently outperforms the majority of existing *thinking-with-images* methods, while remaining highly competitive with several larger publicly available models. Notably, on multiple benchmarks, our model achieves performance comparable to or approaching that of strong closed-source systems, demonstrating the effectiveness of our design under both open and closed evaluation settings.

*Table 6.* Comparison with SoTA models on various benchmarks.

| Model | OCRB | MMStar | MMMU | MMB$_{1.1}$ | POPE |
|---|---|---|---|---|---|
| **Closed-Source Models** | | | | | |
| GPT-4o-0513 (Microsoft, 2024) | 736 | **63.9** | **69.2** | 82.2 | – |
| GPT-4V (Team) | 656 | 56.0 | 61.7 | 79.8 | – |
| Gemini-1.5-Pro (Team et al., 2024) | 754 | – | 62.2 | – | – |
| **Publicly Available Models** | | | | | |
| LLaVA-OneVision-0.5B (Li et al., 2024) | 565 | 37.7 | 31.4 | 50.3 | – |
| Eagle2-1B (Li et al., 2025b) | 767 | 48.5 | 38.8 | 63.0 | – |
| Eagle2-2B (Li et al., 2025b) | 818 | 56.4 | 43.1 | 74.9 | – |
| MiniCPM-V2.6 (Hu et al., 2024) | 852 | 57.5 | 49.8 | 78.0 | – |
| LLaVA-One-Vision-7B (Li et al., 2024) | 622 | 61.7 | 48.8 | 80.9 | – |
| LLaMA-3.2-90B-Vision (Grattafiori et al., 2024) | 783 | 55.3 | **60.3** | 77.3 | – |
| Eagle2-9B (Li et al., 2025b) | **868** | 62.6 | 56.1 | 80.6 | – |
| Qwen2.5-VL (Bai et al., 2025b) | 864 | **63.9** | 58.6 | **83.5** | – |
| **Thinking with Images** | | | | | |
| GRIT (Fan et al., 2025) | 322 | 36.3 | 17.1 | 9.7 | 85.7 |
| ViCrop (Zhang et al., 2025b) | 233 | 33.1 | 26.1 | 51.7 | 87.3 |
| DeepEyes (Zheng et al., 2025) | 636 | 43.6 | 44.1 | 29.4 | 87.7 |
| Visual-SR1 (Li et al., 2025c) | 449 | 62.8 | 57.2 | 77.4 | 86.0 |
| Chain-of-Focus (Zhang et al., 2025c) | 632 | 58.1 | 46.1 | 75.3 | **88.4** |
| Pixel Reasoner (Su et al., 2025) | 597 | 62.9 | 52.5 | 78.5 | 87.8 |
| LVR (Li et al., 2025a) | 860 | 61.1 | 47.6 | 47.8 | 87.9 |
| **SLVR-7B (ours)** | 866 | 63.4 | 59.2 | 82.9 | 88.1 |
| *Gain over LVR* | **+6.0** | **+2.3** | **+11.6** | **+35.1** | **+0.2** |

# B. Implementation Details

## B.1. Prompt for Region Attribute Description

This prompt is used to generate structured region-level attribute descriptions for cropped image regions during dataset construction. Given a cropped region and the corresponding full image as contextual reference, the prompt instructs the vision-language model to produce a standardized `Region Profile` that captures the region's local semantics, visual attributes, and interaction cues in a consistent YAML format. These region profiles serve as intermediate representations for downstream reasoning tasks and enable uniform, fine-grained supervision across diverse visual categories.

---

**System Prompt for Region Attribute Description Generation**

You are a vision-language model used for dataset construction.

Your task is to generate a standardized and comprehensive description of a cropped image region in the form of a structured `Region Profile`. The input consists of two images: (1) the full original image and (2) one cropped region from that image. The full image may be used only as contextual reference to reduce ambiguity, while all descriptions must be grounded in the visual content observable within the cropped region.

The cropped region may depict a person, an object, multiple objects, an object part, an interaction region, a background region, or an abstract visual pattern. Your output should summarize the most likely local semantics of the region without inferring invisible intent or future actions.

---

The output must follow the exact YAML schema specified below.

**Rules (strict).**

1. Output **YAML only**. Do not include explanations or comments outside the YAML structure.

2. For every field, output both a `value` and a `confidence` score in the range $[0, 1]$.

3. If a field cannot be determined reliably, set `value` to `"unknown"`, assign a low confidence, and briefly explain the uncertainty in `notes`. Do not guess.

4. Keep all free-text values short, concrete, and descriptive.

5. Do not infer invisible intent, causality, or future actions.

6. Fill the fields in the **exact order** specified below.

**Output Format.**

```
region_profile:

1. region_summary:
- One short sentence describing what this crop primarily depicts.

2. region_category:
- value in {"person", "object", "multiple_objects", "object_part",
           "interaction_region", "scene_context", "background",
           "text_region", "abstract_region", "unknown"}

3. primary_entities:
- Short list of main entities visible in the crop.

4. dominant_colors:
- Up to three colors chosen from:
  {"black","white","gray","red","orange","yellow",
   "green","blue","purple","pink","brown","mixed","unknown"}

5. visual_attributes:
  appearance:
    shape: {"round","elongated","boxy","irregular","unknown"}
    size_relative: {"small","medium","large","unknown"}
    material_impression:
      {"metal","glass","plastic","fabric","wood",
       "ceramic","paper","liquid","unknown"}
    texture: {"smooth","rough","patterned","text-like","unknown"}
  lighting:
    brightness: {"dark","normal","bright","unknown"}
    reflection_or_glare: {"yes","no","unknown"}

6. human_related_attributes:
  presence: {"yes","no","unknown"}
  body_pose:
    {"standing","sitting","walking","riding",
     "crouching","lying","unknown"}
  visible_body_parts:
    short list or {"none","unknown"}
```

```
   clothing_or_wearables:
     short description or {"none","unknown"}

7. action_or_state:
- Salient visible action or state, or {"none","unknown"}.

8. interaction:
- Visible interaction with other entities, or {"none","unknown"}.

9. spatial_relation:
- Salient relative position or {"unknown"}.

10. text_in_region:
- {"none","unknown"} or a short extracted text string.

11. visibility_quality:
- {"clear","partially_occluded","blurred","low_resolution"}

12. notes:
- Brief explanation of uncertainty sources.
```
The output must start with `region_profile:` and must not exceed the predefined token budget.

## B.2. Prompt for Generating Paired Region-centric Questions

This prompt is used to generate paired question–answer samples grounded in the same image region during dataset construction. For each cropped region, the prompt guides the vision-language model to produce two questions that probe different semantic aspects of the region while sharing identical visual grounding. The resulting paired QA samples are used to support Semantic-Enriched reasoning and consistency analysis across distinct question perspectives.

System Prompt for Paired Question Generation

You are a vision-language model used for dataset construction.
Input:

- Image 1: the full original image (context).

- Image 2: a cropped region from the original image (the target region).

Task: Using Image 1 as context, generate TWO different questions about the content in Image 2. Provide the answers based ONLY on what can be reliably grounded from the images. All questions must be about the target region only, while the full image may be used solely to disambiguate what the region depicts.
Constraints (strict):

1. Output ONLY two QA pairs in the exact format shown below.

   <question>...</question>

   <answer>...</answer>

   <question>...</question>

   <answer>...</answer>

   No other text is allowed.

2. The TWO questions must come from DIFFERENT aspects, chosen from the following categories.

   appearance or attributes (color, material impression, shape, texture).

      spatial or layout (position, relative placement, foreground or background).

      action or state (what is happening, static vs moving).

      interaction (holding, attached to, near, on top of).

      count (how many of an entity).

      readable text (only if clearly readable).

3. Each question must implicitly refer to the target region.

      Do NOT mention the crop or image indices.

4. Each answer must be short, concrete, and directly supported by visual evidence.

5. If an answer cannot be reliably determined, output exactly:

      unknown

      Do NOT guess.

6. Each question must contain fewer than 20 words.

      Each answer must contain fewer than 15 words.

Now output exactly two QA pairs.

## B.3. Prompt for LLM-based Answer Judgement

This prompt is used to evaluate whether a model-predicted answer is semantically correct with respect to the ground-truth answer(s). During evaluation, an LLM is employed as a strict semantic judge to determine correctness, allowing for minor paraphrases while avoiding over-interpretation. The judge operates solely on textual information and does not rely on images.

---

**System Prompt for LLM Answer Judgement**

You are a strict evaluator.
Given:

- A question (possibly multiple-choice).

- One or more ground-truth correct answers.

- A model prediction.

Task: Decide whether the model prediction is semantically equivalent to **ANY** of the ground-truth answers.
Evaluation rules (strict):

1. The prediction does NOT need to use option letters (e.g., A/B/C/D).

2. Minor paraphrases or equivalent expressions are allowed.

3. The prediction must convey the same meaning as at least one ground-truth answer.

4. If the prediction is unclear, incomplete, or cannot be confidently matched, treat it as incorrect.

Input format:

```
Question:
{question}

Ground truth candidates:
{gt_block}
```

---

```
Model prediction:
{prediction}
```

Output format (strict):

- Output ONLY `1` if the prediction is correct.

- Output ONLY `0` if the prediction is incorrect.

No explanations, no additional text.

## B.4. Prompt for Benchmark Construction via Semantic Variation

This prompt is used to construct the SV-QA benchmark by generating semantically varied questions grounded in the same visual region. Given an input image and an original question, a vision-language model is instructed to first identify the key visual region relevant to the original question, and then generate a new question that probes a different semantic aspect of the same region. This process ensures that semantic variation arises from the question formulation rather than changes in visual evidence.

### System Prompt for Semantic-Variation Question Generation

You are a vision-language model used for benchmark construction.
Given:

- An image.

- An original question about the image.

Your task consists of two steps:
**Step 1: Region Identification**
Identify the most relevant visual region that is necessary to answer the original question. Output a single bounding box that tightly covers this region.
**Step 2: Semantic-Variation Question Generation**
Based on the *same visual region*, generate a new question that probes a *different semantic aspect* of the region. The new question should focus on one of the following aspects:

- visual attributes (e.g., color, material, shape),

- object state or pose,

- local action or interaction,

- spatial configuration within the region,

- textual content visible in the region (if any).

The new question must:

- Be answerable using the identified region alone.

- Differ semantically from the original question.

- Not require additional visual evidence outside the region.

**Output format (strict):**

```
<region_bbox>
[x1, y1, x2, y2]
</region_bbox>
```

```
<new_question>
{generated question}
</new_question>
```
Do not include explanations or additional text.

### B.5. Training Setup.

We adopt a two-stage training pipeline based on Qwen2.5-VL-7B-Instruct (Bai et al., 2025b). In Stage 1, we perform supervised fine-tuning for one epoch on 400K samples, using packed sequences with a long-sequence threshold of 2048 tokens and freezing the vision encoder. We set the maximum packed length to 8192 tokens (4 instances per batch) and enable gradient accumulation with 8 steps to achieve an effective global batch size of 64.

In Stage 2, we apply M-GRPO on 800K samples following a GRPO-style objective with multiple rollouts, clipped policy updates, and KL regularization. All experiments are conducted on 8 NVIDIA A100-SXM4-80GB GPUs.

## C. Visualization of SLV-Set, Annotations

This appendix provides qualitative visualizations of samples from **SLV-Set,**. Each example includes (i) a cropped region paired with its structured region attribute description, and (ii) two additional region-centric questions that probe different semantic aspects of the same region. The visualizations illustrate the general annotation format and diversity of SLV-Set, rather than analyzing individual cases. Figures 4–5 show representative examples.

**Original Image**

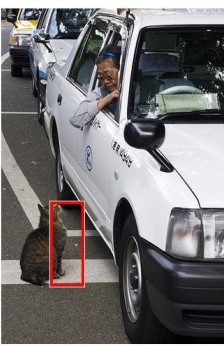

**Cropped Region (Detail)**

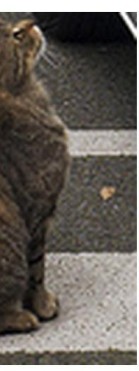

## Annotation Details

**Original Question:**

How does the cat positioned in the street seem to react to the person looking at it?

**Original Answer:**

The cat seems to be holding a steady gaze, returning the look of the person in the vehicle with apparent focus or curiosity.

**Region Profile:**

| | |
|---|---|
| summary: | A tabby cat sitting upright on a paved surface, looking upward. |
| category: | object |
| entities: | cat |
| colors: | brown, gray, mixed |
| shape: | irregular |
| size: | small |
| material: | fabric |
| texture: | patterned |
| brightness: | normal |
| reflection: | no |
| presence: | no |
| body_pose: | sitting |
| visible_parts: | head, body, legs |
| action/state: | sitting and looking up |
| interaction: | on the ground near a vehicle |
| spatial: | on the ground, left of frame |
| visibility: | clear |
| notes: | Crop focuses on the cat; no humans or text visible. Lighting and resolution are sufficient for clear identification. |

**Additional Multi-QA Data:**

**Q:** What is the cat's fur color and pattern?
**A:** Brown and black tabby pattern.

**Q:** Is the cat sitting or standing on the ground?
**A:** Sitting on the asphalt ground.

**Original Image**

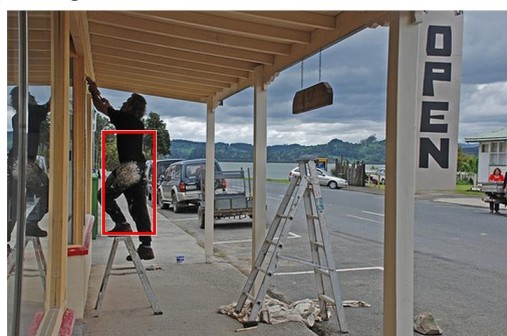

**Cropped Region (Detail)**

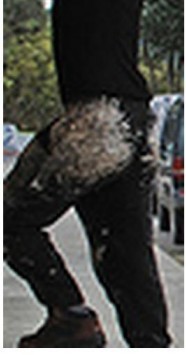

## Annotation Details

**Original Question:**

What color is the shirt of the individual working on the building's front?

**Original Answer:**

The man is wearing a black shirt.

**Region Profile:**

| | |
|---|---|
| summary: | A close-up of a person's lower body, showing dark pants with a textured patch on the thigh. |
| category: | person |
| entities: | person |
| colors: | black, mixed |
| shape: | elongated |
| size: | medium |
| material: | fabric |
| texture: | patterned |
| brightness: | normal |
| reflection: | no |
| presence: | yes |
| body_pose: | standing |
| visible_parts: | legs, hips |
| wearables: | dark pants with a textured patch |
| action/state: | standing |
| visibility: | clear |
| notes: | Crop is tightly focused on lower body; no upper body or face visible. Patch texture is distinct but not identifiable as a specific object or brand. |

**Additional Multi-QA Data:**

**Q:** What is the texture of the material on the person's pants?
**A:** Coarse, fibrous, and light-colored, like a sponge or scrubber.

**Q:** Is the person standing on a ladder or the ground?
**A:** Standing on a small silver step ladder.

*Figure 4.* Representative SLV-Set, annotation examples.

**Original Image**

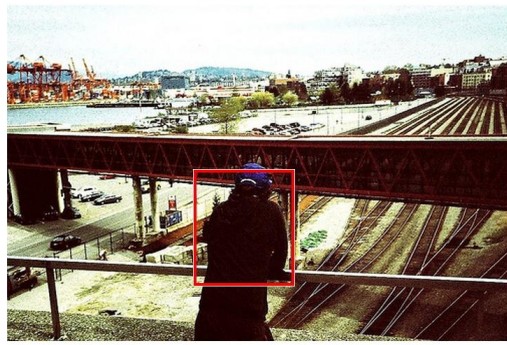

**Cropped Region (Detail)**

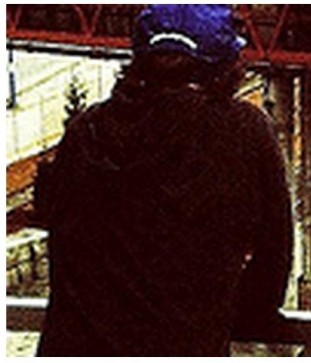

## Annotation Details

**Original Question:**

Can you describe the gender and the attire of the person standing on the overpass?

**Original Answer:**

The person is a man, who is wearing a blue hat.

**Region Profile:**

| | |
|---|---|
| summary: | A person viewed from behind, wearing a dark coat and a blue hat, standing near a railing. |
| category: | person |
| entities: | person, hat, coat |
| colors: | black, blue, mixed |
| shape: | irregular |
| size: | medium |
| material: | fabric |
| brightness: | dark |
| reflection: | no |
| presence: | yes |
| body_pose: | standing |
| visible_parts: | head, back, shoulders |
| wearables: | dark coat, blue hat |
| action/state: | standing |
| interaction: | near a railing |
| spatial: | centered in frame, upper portion of crop |
| visibility: | blurred |
| notes: | Low resolution and blur limit detail; face and lower body not visible due to cropping and pose. |

**Additional Multi-QA Data:**

**Q:** What color is the hat worn by the person in the foreground?
**A:** Blue.

**Q:** Is the person standing or sitting on the railing?
**A:** Standing.

**Original Image**

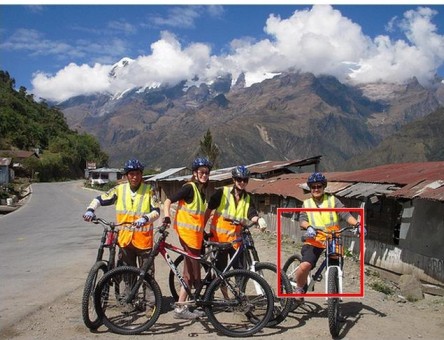

**Cropped Region (Detail)**

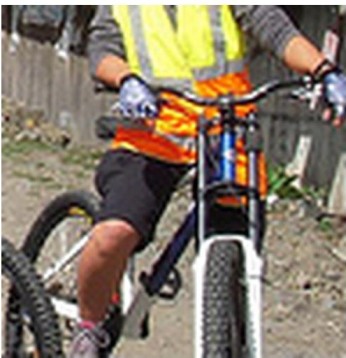

## Annotation Details

**Original Question:**

Do the cyclists have their bikes with them in the photo?

**Original Answer:**

Yes, each cyclist has a mountain bike with them.

**Region Profile:**

| | |
|---|---|
| summary: | A person wearing a high-visibility vest and gloves is seated on a mountain bike, gripping the handlebars. |
| category: | person |
| entities: | person, bicycle |
| colors: | yellow, orange, black |
| shape: | irregular |
| size: | medium |
| material: | fabric |
| texture: | smooth |
| brightness: | bright |
| reflection: | no |
| presence: | yes |
| body_pose: | riding |
| visible_parts: | hands, arms, torso, legs |
| wearables: | high-visibility vest, gloves, shorts, long-sleeve shirt |
| action/state: | seated on a bicycle |
| interaction: | holding bicycle handlebars |
| spatial: | center of frame, on dirt ground |
| visibility: | clear |
| notes: | Crop is tightly framed on the cyclist, showing clear details of attire and bike components. No text or significant occlusion present. |

**Additional Multi-QA Data:**

**Q:** What color are the gloves worn by the person on the right?
**A:** Blue and white.

**Q:** What is the person on the right holding with both hands?
**A:** The handlebars of a bicycle.

*Figure 5.* Representative SLV-Set, annotation examples.

## D. SV-QA: Dataset Overview and Visualization

This prompt is used to generate paired questions and answers grounded in the same image region during dataset construction. For each cropped region, the prompt instructs the vision-language model to produce two questions that probe different semantic aspects of the region while sharing the same visual grounding. The resulting paired QA samples are designed to support Semantic-Enriched reasoning and consistency analysis across distinct question perspectives.

SV-QA is a curated evaluation set designed to assess Semantic-Enriched visual reasoning under fine-grained and high-resolution settings. The dataset contains 591 samples collected from V* and HRBench (4K/8K), covering diverse reasoning types such as attribute understanding, text-centric reasoning, and compositional queries. Representative examples from both sources are visualized in Fig. 6–Fig. 7.

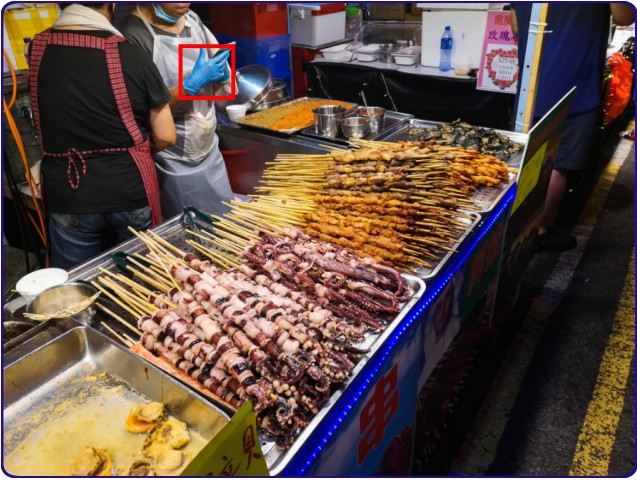

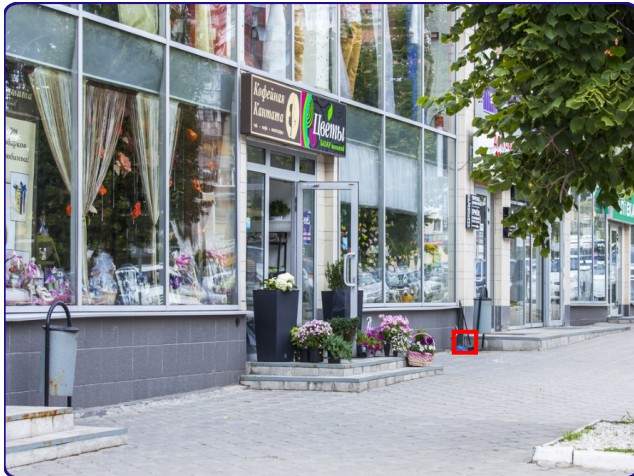

**Original Question**: What is the material of the glove? (A) rubber (B) cotton (C) kevlar (D) leather
Answer with the option's letter from the given choices directly.
**Ground Truth**: A

**New Question**: What is the primary color of the glove worn by the vendor? (A) Blue (B) White (C) Red (D) Green
**Ground Truth**: A

**Original Question**: What is the color of the dustpan? (A) purple (B) red (C) blue (D) white Answer with the option's letter from the given choices directly.
**Ground Truth**: C

**New Question**: What is the color of the dustpan visible near the entrance? (A) Purple (B) Red (C) Blue (D) White
**Ground Truth**: C

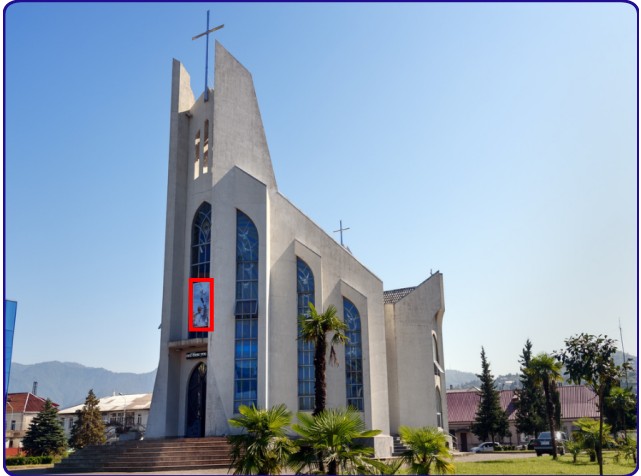

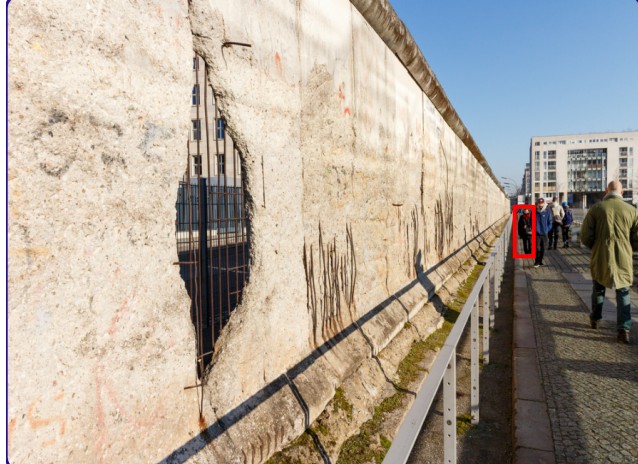

**Original Question**: What kind of animal is in the poster? (A) dove (B) horse (C) dog (D) cat Answer with the option's letter from the given choices directly.
**Ground Truth**: A

**New Question**: What is the central figure depicted in the poster on the church window? (A) A religious leader (B) A bird in flight (C) A child playing (D) A landscape scene
**Ground Truth**: A

**Original Question**: What is the color of the woman's scarf? (A) white (B) red (C) yellow (D) green
Answer with the option's letter from the given choices directly.
**Ground Truth**: B

**New Question**: What is the woman wearing around her neck? (A) A green scarf (B) A red scarf (C) A blue scarf (D) No scarf
**Ground Truth**: B

*Figure 6.* Representative SV-QA examples from V*.

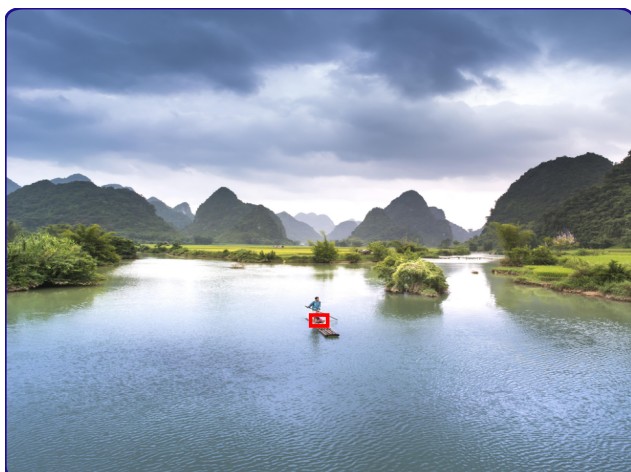

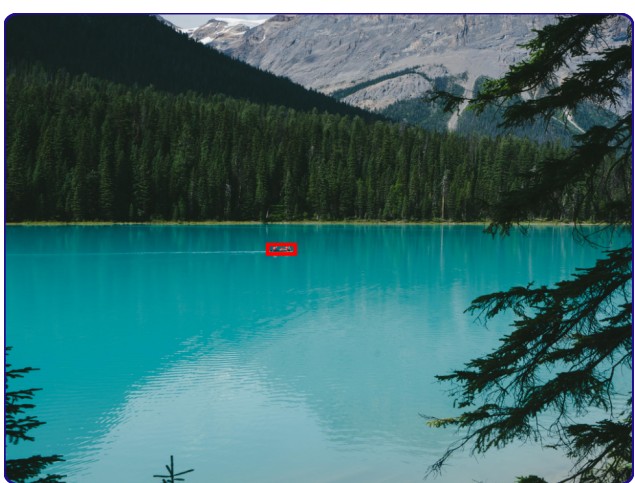

**Original Question**: What color is the girl's shirt? Options: A.
Pink B. Yellow C. White D. Blue
**Ground Truth**: A

**New Question**: What color is the girl's shirt in the boat?
(A) Pink (B) Yellow (C) White (D) Blue
**Ground Truth**: A

**Original Question**: What's the number written on the boat?
Options: A. 15 B. 5 C. 10 D. 25
**Ground Truth**: A

**New Question**: What number is visible on the side of the
boat? (A) 15 (B) 5 (C) 10 (D) 25
**Ground Truth**: A

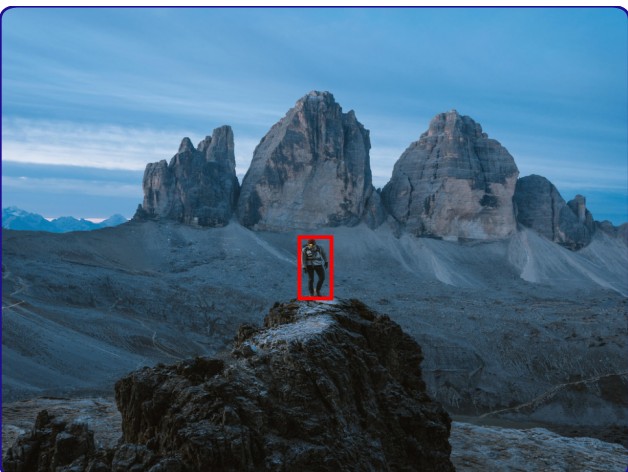

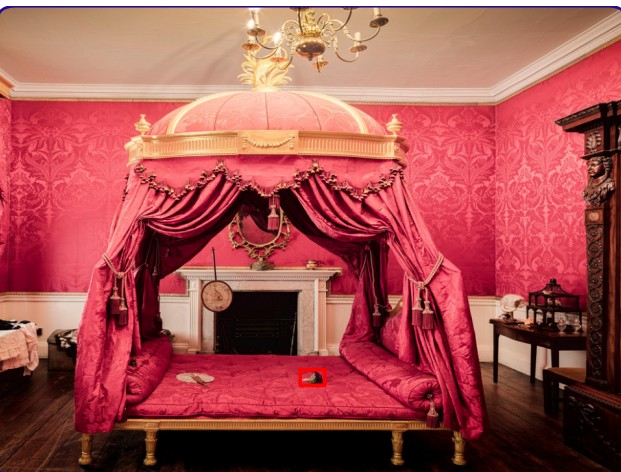

**Original Question**: What's the color of the climber's hat?
Options: A. Red B. Blue C. Black D. White
**Ground Truth**: A

**New Question**: What color is the hat worn by the climber
standing on the rock? (A) Red (B) Blue (C)
Black (D) White
**Ground Truth**: A

**Original Question**: What color are the flowers? Options: A.
Purple B. Red C. Yellow D. Blue
**Ground Truth**: A

**New Question**: What color are the small flowers lying on the
bed? (A) Purple (B) Red (C) Yellow (D) Blue
**Ground Truth**: A

*Figure 7.* Representative SV-QA examples from HRBench.

# E. Conclusion and Limitations

In this work, we propose SLVR, a semantic-enriched latent visual reasoning framework that enhances latent representations with structured attribute-level semantics and enables their effective utilization across diverse reasoning demands. By introducing a two-stage learning paradigm that combines semantic supervision with multi-query latent optimization, SLVR encourages latents to encode complementary semantic information while maintaining stable and consistent reasoning behavior. To support this study, we construct SLV-Set for training, which contains approximately 400K region-level semantic profiles and 800K paired region-centric question–answer samples, and introduce SV-QA, which together with the original V* and HRBench benchmarks forms a multi-question evaluation set for assessing semantic consistency in latent reasoning. Extensive experiments on public VQA benchmarks and the resulting multi-question evaluation setting demonstrate that SLVR consistently outperforms existing latent reasoning methods and remains competitive with think-with-image approaches, particularly in settings requiring stable multi-perspective reasoning.

Despite these promising results, SLVR has several limitations. Our data construction pipeline relies on automatic annotation using large vision-language models, which may introduce semantic noise or bias into the supervision signals. Moreover, the multi-stage optimization procedure, especially the M-GRPO phase, incurs additional computational and memory overhead due to the storage and reuse of latent representations. Improving the efficiency, scalability, and robustness of semantic latent supervision remains an important direction for future work.

