# OpenReview forum: "Semantic-Enriched Latent Visual Reasoning"
_ICML.cc/2026/Conference — ICML 2026 regular_

### Official Review · Reviewer_rykQ · 2026-03-06

**Soundness:** 3
**Presentation:** 3
**Significance:** 2
**Originality:** 2
**Overall Recommendation:** 4
**Confidence:** 2

**Summary:**

The paper proposes SLVR, a two-stage Semantic-Enriched Latent Visual Reasoning framework designed to enhance the semantic depth and robustness of multimodal latent-space reasoning. Beyond traditional thinking with images strategies, SLVR focuses on enriching latent representations with fine-grained, attribute-level semantics. Specifically, in Stage 1, the model learns region-centric latent representations through joint supervision of visual alignment and attribute-level semantic embeddings derived from a Qwen3. In Stage 2, they propose Multi-query Group Relative Policy Optimization (M-GRPO) to align these latent representations across multiple queries grounded in the same region. In addition, they also introduce SV-QA, a benchmark for evaluating latent reasoning under semantic variation.

**Compliance With Llm Reviewing Policy:**

Affirmed.

**Final Justification:**

The author supplemented a large number of experiments to prove the effectiveness of SLVR. Therefore, I can update it to weak accept. However, I hope the author can update the experimental results to the final revision and open the source code to improve reproducibility.

**Key Questions For Authors:**

Please refer to weakness.

**Limitations:**

Yes.

**Strengths And Weaknesses:**

Pos:
1. The proposed M-GRPO adopts multiple query groups for the same regions, breaking the limitation of one change per question and reducing the sharp shift of latent representations.
2. The presentation and teaser are clear, making it easy to understand the main intention and architecture.
3. The authors contribute a new benchmark, SVQA, extending existing visual reasoning benchmarks by querying the same visual region from multiple semantic aspects and reasoning granularities.

Neg:
1. The authors claim that visual supervision is insufficient for capturing “fine-grained semantic structure” and only leads to “appearance-level cues”. It seems somewhat overstated. Recent works in self-supervised visual representation learning have shown that high-level semantics can emerge from visual signals. The paper would be much stronger if the authors provided more experimental results to empirically demonstrate that existing visual-latent methods indeed lack the fine-grained information that SLVR explicitly has.
2. The attribute-level semantic supervision is derived from Qwen3-VL-235B, a more powerful model than the SLVR. This raises a question: is the performance improvement truly due to the semantic-enriched latent design, or is it simply a result of distilling superior world knowledge from a 235B teacher model into a 7B student?
3. The baselines for experimental comparison are too limited. There are many work [1][2] that simultaneously utilize text and visual latent representation without comparison.
[1] Multimodal Chain of Continuous Thought for Latent-Space Reasoning in Vision-Language Models.
[2] Reasoning in the Dark: Interleaved Vision-Text Reasoning in Latent Space.
4. The effectiveness of SV-QA is limited by its small-scale and potential for automatic bias and cyclic evaluation, as its task is confined to simple descriptive attributes that reflect the training objectives. This indicates that the benchmark may have exaggerated performance, as it relies on a single generative model and lacks advanced inference challenges.
5. The experiments mainly focused on models with a parameter scale of 7B. It is not clear whether the gain brought by supervision through additional semantic tokens will marginally diminish on larger parameter models (such as 32B).

If the author can effectively address my concerns, I can consider modifying the score, but at the same time, I will refer to the opinions of other reviewers to make a decision.

---

> ### Author Rebuttal · Authors · 2026-03-31
>
> ### Response to *Visual Supervision vs. Fine-Grained Semantics  (Neg 1)*
>
> We thank the reviewer for this insightful comment. To empirically support this, we add additional evaluations on **VisualPuzzles** and **math benchmarks**:
>
> #### Table 1 VisualPuzzles Results
>
> | Model           | Algorithmic | Analogical | Deductive | Inductive | Spatial | Overall |
> |-----------------|------------:|-----------:|----------:|----------:|--------:|--------:|
> | Qwen2.5-VL      | 35.9        | 26.1       | 35.5      | **28.7**  | 21.3    | 29.2    |
> | LVR             | 31.3        | 25.6       | 40.5      | 24.4      | 26.2    | 29.4    |
> | **SLVR (Ours)** | **37.4**    | **28.0**   | **45.5**  | 26.3      | **33.6**| **34.2**|
>
> #### Table 2 Additional Evaluation on Math Benchmarks
>
> | Dataset     | Qwen2.5-VL | LVR  | SLVR |
> |-------------|-----------:|-----:|-----:|
> | MathVerse   | 45.0       | 39.3 | **45.5** |
> | MathVision  | 25.7       | 21.7 | **27.6** |
> | MathVista   | 68.5       | 67.5 | **68.9** |
>
>
> ### Response to *Teacher Model vs. Semantic-Enriched Latent Design (Neg 2)*
>
> We thank the reviewer for this important question. To examine whether the gain mainly comes from **teacher-data distillation** or from our **semantic-enriched latent design**, we compare several **7B-scale** settings under controlled training pipelines:
>
> - **Qwen2.5-VL-7B + ori data + SFT**: base model trained with the original instruction data.
> - **Qwen2.5-VL-7B + 2q data + SFT**: trained with our **SLV-Set 2q data** only.
> - **Qwen2.5-VL-7B + ori data + SFT + 2q data + GRPO**: further optimized with GRPO on the 2q data.
> -  **SLVR-7B**: our full framework with semantic-enriched latent alignment and multi-query optimization.
>
> #### Table 3 Standard benchmarks
> | Model | OKVQA | GQA | VizWiz | ChartQA | TextVQA | AI2D |  Avg. |
> |------|------:|----:|------:|-------:|--------:|----:|--------------:|
> | Qwen2.5-VL-7B + ori data + SFT | 56.2 | **57.1** | 54.3 | 71.7 | 79.0 | 67.3 | 62.0 |
> | Qwen2.5-VL-7B + 2q data + SFT | 54.7 | 56.8 | 56.8 | 72.5 | **79.3** | 69.4 | 62.3 |
> | Qwen2.5-VL-7B + ori data + SFT + 2q data + GRPO | 54.2 | 56.4 | 56.3 | 73.9 | 79.3 | 71.7 | 62.4 |
> | **SLVR-7B** | **61.8** | 55.6 | **57.8** | **77.2** | 79.3 | **76.0** | **64.1** |
>
> #### Table 4 SV-QA benchmarks
> | Model | V* Q1 | V* Q2 | V* Both | HRBench-4K Q1 | HRBench-4K Q2 | HRBench-4K Both | HRBench-8K Q1 | HRBench-8K Q2 | HRBench-8K Both |
> |------|------:|------:|--------:|--------------:|--------------:|----------------:|--------------:|--------------:|----------------:|
> | Qwen2.5-VL-7B + ori data + SFT | 77.5 | 66.5 | 51.8 | 67.9 | 80.1 | 52.8 | 61.2 | 72.9 | 43.5 |
> | Qwen2.5-VL-7B + 2q data + SFT | 79.6 | 69.6 | 57.1 | 66.5 | 78.4 | 50.1 | 60.5 | 71.1 | 41.1 |
> | Qwen2.5-VL-7B + ori data + SFT + 2q data + GRPO | 80.6 | 75.4 | 58.6 | **70.5** | 82.5 | 55.5 | **63.0** | 72.6 | 47.8 |
> | **SLVR-7B** | **82.2** | **80.1** | **69.1** | 70.4 | **82.8** | **61.1** | 62.5 | **73.6** | **50.6** |
>
> This suggests that the gain does not come solely from distilling stronger teacher knowledge, but from the **joint effect of the constructed data and the proposed latent reasoning design**.
>
> ### Response to *Limited Baseline Comparison (Neg 3)*
>
> We thank the reviewer for providing these relevant references. We carefully checked the suggested works. To the best of our knowledge, **[1]** does not have an available public implementation, so we were unable to include it in a controlled comparison. We therefore additionally compare with **[2]**.
>
> #### Table 5 Comparison with Reasoning in the Dark
>
> | Dataset | Reasoning in the Dark | SLVR-7B |
> |---|---:|---:|
> | AI2D | 60.2 | **76.0** |
> | ChartQA | 5.3 | **77.2** |
> | GQA | 11.6 | **55.6** |
> | OKVQA | 1.1 | **61.8** |
> | TextVQA | 9.2 | **79.3** |
> | VizWiz | 4.2 | **57.8** |
> | HRBench-4K | 44.4 | **70.4** |
> | HRBench-8K | 39.1 | **62.5** |
> | MathVerse | 30.6 | **45.5** |
> | MathVision | 19.2 | **27.6** |
> | MathVista | 50.1 | **68.9** |
> | V*Bench | 41.9 | **69.1** |
>
>
>
> ### Response to *Effectiveness of SV-QA Benchmark  (Neg 4)*
>
> We thank the reviewer for this concern. The SV-QA benchmark in our paper is **manually verified and refined**, which helps reduce noise and potential bias. Moreover, SV-QA is designed to include **multi-perspective and semantically diverse questions for each region**, going beyond simple descriptive attributes and helping reduce the risk of bias or cyclic evaluation tied to specific training objectives.. Relevant examples can be found in **Appendix D (Figures 9–14)**.
>
> We will further expand its scale and diversity in future work.
>
> ### Response to *Unclear Scaling Behavior on Larger Models (e.g., 32B)  (Neg 5)*
>
> We further evaluate **32B-scale models** (Qwen2.5-VL-32B, LVR, and SLVR) on https://anonymous.4open.science/r/icml_rebuttal-39D0/32B_results.md, SLVR continues to achieve consistent improvements over LVR and remains competitive or superior to the original 32B model across multiple benchmarks.

---

> > ### Author Rebuttal · Reviewer_rykQ · 2026-04-03
> >
> > I appreciate the authors' response, but my main concern regarding W2 remains. Standard text-based SFT is a naturally weaker reasoning paradigm due to lower information density compared to the higher-dim latent embeddings used here. The results fail to decouple whether the gain stems from the latent space's superior capacity to absorb high-dimensional knowledge from the 235B oracle or from the SLVR architecture itself. If the same 235B-distilled attribute knowledge were injected into a baseline latent framework (e.g., LVR) without your specific Stage 1 semantic-alignment and Stage 2 GRPO modules, would SLVR still maintain a significant lead?
> >
> > Besides, I will ultimately refer to Reviewer AY3d's comments to decide whether to adjust my score, as the weaknesses raised are also reasonable.

---

> > > ### Author Response · Authors · 2026-04-05
> > >
> > > ### Response to Latent Capacity vs. Method Contribution
> > > We thank the reviewer for the insightful question. We address this concern from two perspectives.
> > >
> > > First, regarding whether the improvement comes from latent capacity or our method design.
> > > We agree that latent representations are inherently more information-dense than text-based SFT. However, existing latent-based methods (e.g., LVR) lack explicit supervision on how high-dimensional semantic information should be organized in the latent space. Our key contribution is not simply leveraging latent capacity, but providing a structured supervision signal for latent reasoning. In Stage 1, we introduce semantic-aligned supervision to guide the model to encode region-level information into a structured latent space; in Stage 2, RL further optimizes this representation toward task-relevant reasoning. Therefore, the gain is not from latent capacity alone, but from how the latent space is structured and optimized.
> > >
> > > Second, to directly test whether injecting the same 235B-distilled knowledge into a baseline latent framework is sufficient, we construct an additional experiment. At inference time, we use the attribute outputs from Qwen3-VL-235B as semantic prompts, which are injected into the latent reasoning process of LVR (denoted as “LVR + Semantic inject”). This setting provides the same high-dimensional semantic knowledge to LVR without our Stage 1 semantic alignment or Stage 2 GRPO design.
> > >
> > > The results are shown below:
> > >
> > > ### Standard Benchmarks
> > >
> > > | Model | OKVQA | GQA | VizWiz | ChartQA | TextVQA | AI2D |
> > > |------|------:|----:|------:|-------:|--------:|----:|
> > > | LVR | 50.6 | **57.4** | 33.1 | 64.4 | 75.1 | **77.3** |
> > > | LVR + Semantic inject | 59.5 | 55.1 | 36.5 | 67.0 | 76.4 | 75.6 |
> > > | SLVR-7B | **61.8** | 55.6 | **57.8** | **77.2** | **79.3** | 76.0 |
> > >
> > > ### SV-QA
> > >
> > > | Model | V* Q1 | V* Q2 | V* Both | HRBench-4K Q1 | HRBench-4K Q2 | HRBench-4K Both | HRBench-8K Q1 | HRBench-8K Q2 | HRBench-8K Both |
> > > |------|------:|------:|--------:|--------------:|--------------:|----------------:|--------------:|--------------:|----------------:|
> > > | LVR | 81.7 | 77.5 | 65.4 | 70.1 | 80.4 | 57.9 | 62.9 | 71.1 | 46.6 |
> > > | LVR + Semantic inject | 81.1 | 79.1 | 63.9 | 69.9 | 78.6 | 59.0 | **63.2** | 72.0 | 48.4 |
> > > | SLVR-7B | **82.2** | **80.1** | **69.1** | **70.4** | **82.8** | **61.1** | 62.5 | **73.6** | **50.6** |
> > >
> > > We observe that directly injecting the same semantic knowledge into LVR brings only limited improvements and does not match SLVR. In particular, gains on **V* Both** and **HRBench Both** remain significantly lower, indicating weaker cross-query consistency. If the improvement were mainly due to latent capacity or high-dimensional knowledge injection, this baseline should achieve comparable performance. However, this is not observed.
> > >
> > > ### Response to Related Concerns (Reviewer AY3d)
> > >
> > > We sincerely thank the reviewer for the detailed feedback and for carefully examining our rebuttal. We also appreciate the reviewer’s note about considering Reviewer AY3d’s comments, and we hope the following clarifications further address the remaining concerns.
> > >
> > >
> > > Regarding the concern that the semantic design may be too rigid or lossy, our main point is that the template should be understood within the two-stage training strategy. Stage 1 provides structured supervision to teach the model to enter the latent reasoning mode within `<sem_start>` and `<sem_end>`, while Stage 2 further optimizes the latent toward what is most useful for downstream reasoning via RL and multi-query consistency. Thus, the final latent is not fixed by the template, but refined toward richer semantics and stronger reasoning behavior.
> > >
> > > To directly test whether the template is restrictive, we compare SLVR-7B with a controlled baseline, **LVR + Multi-Q**, which is directly optimized with GRPO on the same 2q data but does not learn the semantic template in Stage 1.
> > >
> > >
> > > | Model | OKVQA | GQA | VizWiz | ChartQA | TextVQA | AI2D |Avg|
> > > |------|------:|----:|------:|-------:|--------:|----:|-----:|
> > > | LVR + Multi-Q | 49.9 | 54.5 | 57.4 | 69.0 | 72.3 | **76.2** | 63.2 |
> > > | SLVR-7B | **61.8** | **55.6** | **57.8** | **77.2** | **79.3** | 76.0 | **68.0**|
> > >
> > >
> > > | Model | V* Q1 | V* Q2 | V* Both | HRBench-4K Q1 | HRBench-4K Q2 | HRBench-4K Both | HRBench-8K Q1 | HRBench-8K Q2 | HRBench-8K Both |
> > > |------|------:|------:|--------:|--------------:|--------------:|----------------:|--------------:|--------------:|----------------:|
> > > | LVR + Multi-Q | 80.6 | 77.5 | 65.4 | 68.4 | 81.4 | 58.6 | 60.4 | 71.4 | 49.7 |
> > > | SLVR-7B | **82.2** | **80.1** | **69.1** | **70.4** | **82.8** | **61.1** | **62.5** | **73.6** | **50.6** |
> > >
> > >
> > > Due to space limitations, we provide additional results on more complex reasoning:
> > > https://anonymous.4open.science/r/icml_rebuttal-39D0/visualpuzzles_math2.md

---

### Official Review · Reviewer_ZJBw · 2026-03-09

**Soundness:** 2
**Presentation:** 2
**Significance:** 2
**Originality:** 2
**Overall Recommendation:** 5
**Confidence:** 3

**Summary:**

This paper focuses on multimodal latent-space reasoning. Existing methods rely heavily on visual supervision and often produce latent representations lacking sufficient semantic information. To address these issues, the paper proposes a two-stage learning framework called Semantic-Enriched Latent Visual Reasoning (SLVR), which learns fine-grained alignment between image regions and semantic attributes within the latent representation space. In addition, the authors introduce SLV-Set and SV-QA, which define a more challenging task setting where two questions from different perspectives are associated with the same image region.

**Compliance With Llm Reviewing Policy:**

Affirmed.

**Final Justification:**

The authors addressed my previous questions during the rebuttal. The experiments are solid, the presentation is clear, and the work makes a significant contribution to the field. There are no major issues, so I give a final score of 5.

**Key Questions For Authors:**

At present, I can only give a score of 3. However, if the following questions can be addressed, I may raise my score to up to 5:

1. If the paper clearly states that SLV-set and SV-QA will be fully released in the near future, I will increase my score to 4.

2. If all the models evaluated on the SV-QA benchmark are trained on SLV-set—meaning that Table 2 presents a fair comparison—and Table 1 is supplemented with experiments on additional datasets showing that SLVR achieves improvements on most datasets, I will increase my score to 5.

Note that if only the second point is satisfied but the first point is not, I will still keep the score at 3, since the first point is more important.

**Limitations:**

yes

**Strengths And Weaknesses:**

This paper proposes a new benchmark that is more complex and challenging than existing ones, with broad potential impact and generally reliable contributions. However, there are still some shortcomings.

First, since the authors have already constructed a more complex dataset, it is unclear why the number of questions for each image region is restricted to two. If the number of questions were variable and greater than or equal to two, evaluating the probability that all questions are answered correctly might provide a more meaningful metric.

Second, the number of models evaluated on SV-QA is relatively limited. It would be better to include all the models listed in Table 1 in these experiments. A good benchmark should be supported by rich and comprehensive experimental evaluations to demonstrate its difficulty and value.

Regarding the other methodological contribution of this paper, I have two concerns about its technical soundness. First, the paper claims that visual and semantic information are more effectively integrated into the latent representation. However, why does the model perform worse than the latent representation without semantic information on the GQA and AI2D datasets?

Second, for the experiments on the SV-QA dataset, were the compared methods also trained on SLV-set to ensure a fair comparison? I could not find a description of this in the paper. If such details exist, please indicate where they are discussed.

---

> ### Author Rebuttal · Authors · 2026-03-29
>
> ### Response to *Number of Questions per Region (First Weakness)*
>
> We thank the reviewer for this insightful suggestion. The current design with **two questions per region (2q)** is chosen as a **cost-effective trade-off** between data diversity and training efficiency.  At the same time, it already enables meaningful multi-perspective reasoning and consistency learning.
>
>
> We agree that extending to **a variable number of questions (≥2)** and evaluating stricter metrics (e.g., all-correct probability) is promising, and we plan to explore this direction by scaling up the dataset in future work.
>
> ### Response to *Limited Number of Models on SV-QA (Second Weakness)*
>
> We thank the reviewer for this valuable suggestion. Due to space limitation, we include all the models listed in Table 1 in these experiments in https://anonymous.4open.science/r/icml_rebuttal-39D0/sv-qa_results.md
>
> ### Response to *Performance on GQA and AI2D  (First concern)*
>
> We thank the reviewer for this question. The relatively weaker performance on GQA and AI2D is mainly due to their dataset characteristics. GQA requires complex global compositional reasoning with long dependency chains, while AI2D emphasizes diagram understanding and symbolic relational structures. These tasks rely more on holistic reasoning and abstract representations, which are less aligned with our current region-level semantic modeling.
>
> In future work, we will focus on improving the integration of global reasoning and symbolic understanding within the latent representation.
>
> ### Response to *Fairness of Training on SLV-Set (Second concern)*
>
> We thank the reviewer for this suggestion. We will clarify this point in the **Response to Key Questions**.
>
> ### Response to *Dataset Release (Key Question 1)*
>
> Yes, we will fully release **SLV-Set** and **SV-QA** upon acceptance.
>
> As stated in the contribution section of the introduction, both datasets are key components of our work. We will open-source them to support the community and facilitate further research on multimodal reasoning.
>
> ### Response to *Fair Comparison on SV-QA  (Key Question 2)*
>
> We thank the reviewer for this important suggestion. To ensure a fair comparison, we add the following experiments under **the same training settings as described below**.
>
> - **Qwen2.5-VL-7B + ori data + SFT**: the base model trained with the original instruction data using supervised fine-tuning.
> - **Qwen2.5-VL-7B + 2q data + SFT**: the model further trained on our **SLV-Set 2q data**.
> - **Qwen2.5-VL-7B + ori data + SFT + 2q data + GRPO**: the model additionally optimized with GRPO.
> - **SLVR-7B**: our full model.
>
> - **Table 1 (SV-QA benchmarks)**: all models are evaluated under the same protocol.
> - **Table 2 (Standard benchmarks)**: we further evaluate these aligned variants on public datasets to verify generalization.
>
> #### Table 1 SV-QA benchmarks
> | Model | V* Q1 | V* Q2 | V* Both | HRBench-4K Q1 | HRBench-4K Q2 | HRBench-4K Both | HRBench-8K Q1 | HRBench-8K Q2 | HRBench-8K Both |
> |------|------:|------:|--------:|--------------:|--------------:|----------------:|--------------:|--------------:|----------------:|
> | Qwen2.5-VL-7B + ori data + SFT | 77.5 | 66.5 | 51.8 | 67.9 | 80.1 | 52.8 | 61.2 | 72.9 | 43.5 |
> | Qwen2.5-VL-7B + 2q data + SFT | 79.6 | 69.6 | 57.1 | 66.5 | 78.4 | 50.1 | 60.5 | 71.1 | 41.1 |
> | Qwen2.5-VL-7B + ori data + SFT + 2q data + GRPO | 80.6 | 75.4 | 58.6 | **70.5** | 82.5 | 55.5 | **63.0** | 72.6 | 47.8 |
> | **SLVR-7B** | **82.2** | **80.1** | **69.1** | 70.4 | **82.8** | **61.1** | 62.5 | **73.6** | **50.6** |
>
>
> #### Table 2 Standard benchmarks
> | Model | OKVQA | GQA | VizWiz | ChartQA | TextVQA | AI2D | Avg. |
> |------|------:|----:|------:|-------:|--------:|----:|------:|
> | Qwen2.5-VL-7B + ori data + SFT | 56.2 | **57.1** | 54.3 | 71.7 | 79.0 | 67.3 | 62.0 |
> | Qwen2.5-VL-7B + 2q data + SFT | 54.7 | 56.8 | 56.8 | 72.5 | **79.3** | 69.4 | 62.3 |
> | Qwen2.5-VL-7B + ori data + SFT + 2q data + GRPO | 54.2 | 56.4 | 56.3 | 73.9 | 79.3 | 71.7 | 62.4 |
> | **SLVR-7B** | **61.8** | 55.6 | **57.8** | **77.2** | 79.3 | **76.0** | **64.1** |
>
> These comparisons will be included in the revised version to clearly demonstrate fairness.

---

> > ### Author Rebuttal · Reviewer_ZJBw · 2026-04-01
> >
> > The authors may have somewhat misunderstood my question2. Since Table 1 in the paper includes six datasets, and the proposed method is not the best on two of them, I suggested conducting experiments on additional datasets. However, the authors only added experiments for Qwen2.5-VL-7B under an additional model setting, rather than evaluating on extra datasets.

---

> > > ### Author Response · Authors · 2026-04-02
> > >
> > > ### Response to *Additional Datasets Evaluation (Question 2)*
> > >
> > > We thank the reviewer for the clarification and for the suggestion. We further include additional evaluations on more datasets, as shown in **Table 5** in the Appendix A, covering broader comparisons.
> > >
> > > ####  Additional Datasets Evaluation (from Table 5)
> > >
> > > | Model | OCRB | MMStar | MMMU | MMB₁.₁ | POPE | Avg |
> > > |---|---:|---:|---:|---:|---:|---:|
> > > | GRIT | 322 | 36.3 | 17.1 | 9.7 | 85.7 | 36.2 |
> > > | ViCrop | 233 | 33.1 | 26.1 | 51.7 | 87.3 | 44.3 |
> > > | DeepEyes | 636 | 43.6 | 44.1 | 29.4 | 87.7 | 53.7 |
> > > | Visual-SR1 | 449 | 62.8 | 57.2 | 77.4 | 86.0 | 65.7 |
> > > | Chain-of-Focus | 632 | 58.1 | 46.1 | 75.3 | 88.4 | 66.2 |
> > > | Pixel Reasoner | 597 | 62.9 | 52.5 | 78.5 | 87.8 | 68.3 |
> > > | LVR | 860 | 61.1 | 47.6 | 47.8 | 87.9 | 66.1 |
> > > | **SLVR-7B (Ours)** | **866** | **63.4** | **59.2** | **82.9** | 88.1 | **76.4** |
> > >
> > > Across these **5 datasets**, our method achieves the **best performance on 4 out of 5 benchmarks**, and is only slightly below the best on POPE. The **overall average performance is also the highest** (SLVR: **76.4**, higher than all compared methods).
> > >
> > > In addition, we further evaluate on datasets requiring more **abstract reasoning**:
> > >
> > > #### Table 1 VisualPuzzles Results
> > >
> > > | Model | Algorithmic | Analogical | Deductive | Inductive | Spatial | Overall |
> > > |------|------------:|-----------:|----------:|----------:|--------:|--------:|
> > > | Qwen2.5-VL | 35.9 | 26.1 | 35.5 | **28.7** | 21.3 | 29.2 |
> > > | LVR | 31.3 | 25.6 | 40.5 | 24.4 | 26.2 | 29.4 |
> > > | **SLVR (Ours)** | **37.4** | **28.0** | **45.5** | 26.3 | **33.6** | **34.2** |
> > >
> > > #### Table 2 Math Benchmarks Results
> > >
> > > | Dataset | Qwen2.5-VL | LVR | SLVR |
> > > |--------|-----------:|----:|-----:|
> > > | MathVerse | 45.0 | 39.3 | **45.5** |
> > > | MathVision | 25.7 | 21.7 | **27.6** |
> > > | MathVista | 68.5 | 67.5 | **68.9** |
> > >
> > > These additional results further demonstrate that our method consistently improves performance across **diverse datasets and reasoning types**, beyond the original six benchmarks.

---

### Official Review · Reviewer_v2zU · 2026-03-11

**Soundness:** 3
**Presentation:** 3
**Significance:** 3
**Originality:** 3
**Overall Recommendation:** 4
**Confidence:** 2

**Summary:**

This paper introduces Semantic-Enriched Latent Visual Reasoning (SLVR), a novel two-stage framework designed to enhance latent-space reasoning in Multimodal Large Language Models (MLLMs) by enriching compact visual representations with fine-grained semantics. To overcome the limitations of purely visually-supervised latents and computationally expensive explicit visual cropping, SLVR first employs structured latent learning to align a dedicated semantic token with detailed region profiles, followed by an RL-based Multi-query Group Relative Policy Optimization (M-GRPO) to enforce cross-query consistency and prevent semantic drift. Supported by a newly constructed training dataset (SLV-Set) and a robustness evaluation benchmark (SV-QA), the authors demonstrate that explicitly integrating attribute-level supervision with multi-query alignment significantly improves reasoning accuracy and consistency, allowing SLVR to consistently outperform existing latent reasoning baselines and achieve performance highly competitive with heavier explicit visual reasoning paradigms.

**Compliance With Llm Reviewing Policy:**

Affirmed.

**Key Questions For Authors:**

1. Could the authors provide a quantitative analysis of the training time and memory overhead introduced by the M-GRPO stage compared to standard supervised fine-tuning? Additionally, how sensitive is the model's performance to the RL-specific hyperparameters in Stage 2?

2. Given the "black-box" nature of latent reasoning, can the authors provide further qualitative visualizations (e.g., cross-attention maps or t-SNE projections of the semantic tokens) to explicitly demonstrate that the latents are effectively capturing and disentangling the fine-grained attributes as claimed?

3. Since the structured latent learning in Stage 1 relies heavily on the explicit attribute supervision from the SLV-Set, how does the SLVR framework perform when encountering out-of-distribution (OOD) objects, rare attributes, or complex spatial relations not covered in the training distribution?

4. How does the performance of the SLVR framework scale with the size of the backbone MLLM?

**Limitations:**

yes

**Strengths And Weaknesses:**

**Strengths:**

*  The paper effectively identifies and addresses a critical bottleneck in MLLMs: the trade-off between the computational inefficiency of explicit visual cropping and the lack of fine-grained semantic understanding in standard latent representations.
*   The two-stage framework is highly logical. Using explicit attribute supervision to ground the "semantic latent" token (Stage 1), followed by the novel M-GRPO algorithm to enforce cross-query consistency (Stage 2), is a technically sound and creative approach.
*   The introduction of the large-scale SLV-Set and the targeted SV-QA benchmark are significant contributions to the community, providing both the necessary training data and a rigorous evaluation protocol for semantic robustness.

**Weaknesses:**

*   The two-stage training pipeline, particularly the integration of RL-based optimization (M-GRPO), likely introduces substantial training complexity, computational overhead, and sensitivity to hyperparameters compared to standard supervised fine-tuning.
*   Latent reasoning is inherently a "black box." The paper would be significantly stronger if it included more in-depth visualizations (e.g., t-SNE clusters of the semantic tokens or cross-attention maps) to explicitly prove that the latent tokens are capturing the claimed fine-grained attributes.
*  Because Stage 1 relies heavily on explicit attribute supervision from the constructed SLV-Set, it is unclear how well the learned semantic latents generalize in a zero-shot manner to entirely unseen, rare, or highly complex attributes outside the training distribution.

---

> ### Author Rebuttal · Authors · 2026-03-31
>
> ### Response to *Training Complexity and Computational Overhead (Weakness 1 and Key Question 1)*
>
> We thank the reviewer for raising this important question.We note that **SFT + GRPO is already a mainstream training paradigm** for multimodal large models. The additional complexity in our method mainly comes from the **multi-query RL stage**. Under the same setting (**8×A800 80GB**, same data scale). The results are as follows:
>
> | Setting | total_flos | train_runtime | train_samples_per_second | train_steps_per_second |
> |---|---:|---:|---:|---:|
> | GRPO (single-query) | 1.0608e20 | 402,145.803 s (~111.7 h) | 2.007 | 0.00807 |
> | GRPO (2q, independent) | 1.4232e20 | 539,524.804 s (~149.9 h) | 1.496 | 0.00602 |
> | M-GRPO (ours, joint) | 1.9635e20 | 744,344.405 s (~206.8 h) | 1.084 | 0.00436 |
>
> While M-GRPO introduces additional computational overhead compared to single-query GRPO, it brings clear and consistent performance gains, as demonstrated in the ablation studies (Table 3 in the main paper), where **GRPO (Single-Q)**, **Multi-Q**, and **SLVR-7B** show progressively improved results.
>
>
> ### Response to *Sensitivity Analysis of Stage 2 RL Hyperparameters (Weakness 2 and Key Question 2)*
>
> We further evaluate the sensitivity of Stage 2 RL hyperparameters on **SV-QA** using three different reward weight settings:
>
> - **Setting A**:
>   λ_ans = 1.0, λ_cons = 0.5, λ_stab = 0.2
>
> - **Setting B**:
>   λ_ans = 1.0, λ_cons = 0.5, λ_stab = 0.3
>
> - **Setting C (Ours)**:
>   λ_ans = 1.0, λ_cons = 0.5, λ_stab = 0.4
>
> | Setting | λ_ans | λ_cons | λ_stab | V* Q1 | V* Q2 | V* Both | HRBench-4K Q1 | HRBench-4K Q2 | HRBench-4K Both | HRBench-8K Q1 | HRBench-8K Q2 | HRBench-8K Both |
> |---|---:|---:|---:|---:|---:|---:|---:|---:|---:|---:|---:|---:|
> | A | 1.0 | 0.5 | 0.2 | 81.7 | 79.1 | 67.5 | 67.8 | 80.4 | 59.8 | 62.0 | 72.5 | 48.9 |
> | B | 1.0 | 0.5 | 0.3 | **82.2** | 79.6 | 68.6 | 70.1 | 82.4 | 60.7 | 62.3 | **73.8** | 50.4 |
> | C (Ours) | 1.0 | 0.5 | 0.4 | **82.2** | **80.1** | **69.1** | **70.4** | **82.8** | **61.1** | **62.5** | 73.6 | **50.6** |
>
> **Conclusion.**
> The performance is relatively stable across different hyperparameter choices, while our default setting achieves the best overall balance.
>
>
> ### Response to *Black-box Nature of Latent Reasoning (Weakness 2 and Key Question 2)*
>
> We provide a qualitative visualization using **t-SNE projections** of different token types (text, vision, LVR token, and our semantic token), as shown in:
> https://anonymous.4open.science/r/icml_rebuttal-39D0/t-SNE.png. The visualization is generated from **100 randomly sampled test instances**.
>
> The results indicate that our semantic tokens exhibit clearer structure and better alignment in the multimodal space.
>
> Additional qualitative visualizations are provided at:
> https://anonymous.4open.science/r/icml_rebuttal-39D0/example1.png to
> https://anonymous.4open.science/r/icml_rebuttal-39D0/exampl8.png
>
> #### Table 1 VisualPuzzles Results
>
> | Model           | Algorithmic | Analogical | Deductive | Inductive | Spatial | Overall |
> |-----------------|------------:|-----------:|----------:|----------:|--------:|--------:|
> | Qwen2.5-VL      | 35.9        | 26.1       | 35.5      | **28.7**  | 21.3    | 29.2    |
> | LVR             | 31.3        | 25.6       | 40.5      | 24.4      | 26.2    | 29.4    |
> | **SLVR (Ours)** | **37.4**    | **28.0**   | **45.5**  | 26.3      | **33.6**| **34.2**|
>
> #### Table 2  Math Benchmarks results
>
> | Dataset     | Qwen2.5-VL | LVR  | SLVR |
> |-------------|-----------:|-----:|-----:|
> | MathVerse   | 45.0       | 39.3 | **45.5** |
> | MathVision  | 25.7       | 21.7 | **27.6** |
> | MathVista   | 68.5       | 67.5 | **68.9** |
>
> These results further supporting that our semantic tokens effectively capture structured and disentangled information.
>
> ### Response to *Generalization to OOD Objects and Attributes (Weakness 3 and Key Question 3)*
>
> We note that **V\*** and **HRBench (4K/8K)** in Table 2 are already OOD benchmarks, and we further evaluate on additional OOD datasets in Table 1 (VisualPuzzles, including spatial reasoning) and Table 2 (math benchmarks). The consistent improvements across these settings demonstrate that SLVR generalizes well to unseen objects, attributes, and complex relations.
>
>
>
> ### Response to *Scaling to Larger Models (Key Question 4)*
>
> We further evaluate **32B-scale models** (Qwen2.5-VL-32B, LVR, and SLVR) on:
> https://anonymous.4open.science/r/icml_rebuttal-39D0/32B_results.md. SLVR continues to achieve consistent improvements over LVR and remains competitive or superior to the original 32B model across multiple benchmarks.
>
> For this setting, all models are trained on the **same dataset** and evaluated under **aligned conditions**. Due to computational constraints, we focus on the **SFT stage** to provide a controlled and efficient validation. All experiments are conducted on **8×A800 80GB GPUs**.

---

> > ### Author Rebuttal · Reviewer_v2zU · 2026-04-04
> >
> > Thanks for the response. All my concerns are resolved. Please include these additional discussion and experiments in you revision.

---

> > > ### Author Response · Authors · 2026-04-04
> > >
> > > Thank you for your positive feedback and for recognizing our efforts. We are glad that your concerns have been addressed. We will incorporate the additional discussions and experimental results into the revised version of the paper.

---

### Official Review · Reviewer_AY3d · 2026-03-13

**Soundness:** 2
**Presentation:** 3
**Significance:** 3
**Originality:** 2
**Overall Recommendation:** 4
**Confidence:** 4

**Summary:**

The paper proposes SLVR for latent visual reasoning. It first constructs a dataset by using Qwen3-VL-235B to obtain region-level semantic annotations and region-centric question-answer pairs (where two questions are generated for one visual region), along with a small eval benchmark called SV-QA. Then, it has two stage of training where stage 1 aligns region latents to the visual features and semantic latents to attribute embeddings, while stage 2 performs multi-query GRPO with rewards on answer correctess, latent consistency and stability.

**Compliance With Llm Reviewing Policy:**

Affirmed.

**Final Justification:**

After reading the authors’ rebuttal and follow-up discussion, I think the paper improved substantially through the additional experiments and clarifications, and these should be incorporated into the revision. I still have reservations regarding nontrivial amount of annotation error, not very strong gains on HRBench, and inherent limitations in flexibility and expressiveness of template-based design. That said, I appreciate the authors’ honesty in reporting these numbers and clarifying the annotation process. Since a human evaluation has already been conducted to flag errors, I strongly encourage the authors to correct the dataset before release; and/or, an automatic filtering or correction pipeline could further improve the quality of the annotations. Overall, I view this as a borderline paper and remain open to discussion with the other reviewers and the AC, but I am currently leaning toward a weak accept.

**Key Questions For Authors:**

- Have you validated the quality of the Qwen generated region semantic profiles and QA pairs with human evaluation? How often are the generated attributes incomplete, hallucinated, or inconsistent with the image?
- Have you performed ablations over attribute types to determine which semantic fields are actually useful?
- Why are the proposed methods worse on benchmarks in Table 5? Could you provide some failure case analysis?

**Limitations:**

yes

**Strengths And Weaknesses:**

### Strengths
- The paper is clearly written and easy to follow. The visualization is of high quality.
- The introduced benchmarks have an abundance annotations that might benefit the community.
- It has a good coverage of visual reasoning benchmarks, but should include major results from Appendix A to main paper

### Weakness
- Heavy reliance on synthetic supervision and evaluation. The attribute profiles, multi-query training pairs, and SV-QA are all generated with Qwen3-VL-235B, and answer correctness in stage2 is judged by another external Qwen model (how's the accuracy?). The quality of Qwen3-VL-235B generated annotations are unclear. Besides, I'm curious to see the zero-shot performance of Qwen3-VL-235B on visual reasoning tasks.
- Concerns on semantics attributes: (i) The semantic attribute schema is fixed and manually templated, which may work for simple object-centric questions, but is less likely to capture richer semantics; (ii) All attributes are collapsed into a single 4096-dimensional embedding, which is likely too lossy. It's very likely that the question might not focus on one attribute, while the rest are redundant or irrelevant.
- The paper uses a similar framework as LVR, but the performance gain is not very strong compared to LVR and pretrained MLLMs on public benchmarks (Table 1 and Table 5), raising doubts on the originality and effectiveness.

---

> ### Author Rebuttal · Authors · 2026-03-28
>
> ### Response to *Heavy Reliance on Synthetic Supervision (Weakness 1 and Key Questions 1)*
>
> To examine whether the gains mainly come from **synthetic data** or the **combination of data and our method**, we conduct controlled experiments using Qwen2.5-VL-7B under different training settings.
>
> #### Table 1 Standard benchmarks
> | Model | OKVQA | GQA | VizWiz | ChartQA | TextVQA | AI2D |  Avg. |
> |------|------:|----:|------:|-------:|--------:|----:|--------------:|
> | Qwen2.5-VL-7B + ori data + SFT | 56.2 | **57.1** | 54.3 | 71.7 | 79.0 | 67.3 | 62.0 |
> | Qwen2.5-VL-7B + 2q data + SFT | 54.7 | 56.8 | 56.8 | 72.5 | **79.3** | 69.4 | 62.3 |
> | Qwen2.5-VL-7B + ori data + SFT + 2q data + GRPO | 54.2 | 56.4 | 56.3 | 73.9 | 79.3 | 71.7 | 62.4 |
> | **SLVR-7B** | **61.8** | 55.6 | **57.8** | **77.2** | 79.3 | **76.0** | **64.1** |
>
> #### Table 2 SV-QA benchmarks
> | Model | V* Q1 | V* Q2 | V* Both | HRBench-4K Q1 | HRBench-4K Q2 | HRBench-4K Both | HRBench-8K Q1 | HRBench-8K Q2 | HRBench-8K Both |
> |------|------:|------:|--------:|--------------:|--------------:|----------------:|--------------:|--------------:|----------------:|
> | Qwen2.5-VL-7B + ori data + SFT | 77.5 | 66.5 | 51.8 | 67.9 | 80.1 | 52.8 | 61.2 | 72.9 | 43.5 |
> | Qwen2.5-VL-7B + 2q data + SFT | 79.6 | 69.6 | 57.1 | 66.5 | 78.4 | 50.1 | 60.5 | 71.1 | 41.1 |
> | Qwen2.5-VL-7B + ori data + SFT + 2q data + GRPO | 80.6 | 75.4 | 58.6 | **70.5** | 82.5 | 55.5 | **63.0** | 72.6 | 47.8 |
> | **SLVR-7B** | **82.2** | **80.1** | **69.1** | 70.4 | **82.8** | **61.1** | 62.5 | **73.6** | **50.6** |
>
> The proposed **2q data**—i.e., the two-question data constructed in our **SLV-Set**.  The best results are achieved only when combining this data with our full framework, indicating that the gains come from the **joint effect of data and algorithm**.
>
>
> ### Response to *Unverified Reliability of External Reward Model (Weakness 1)*
>
> We note that using **LLMs as reward judges** has become a common practice in recent GRPO-based works (e.g., Deepeyes, Visual-SR1) . In our setting, Qwen3-max serves as a sufficiently powerful judge to provide stable and informative supervision signals. The overall performance improvement demonstrates the acceptability of the judge model's accuracy.
>
>
> ### Response to *Unclear Quality of Automatically Generated Annotations (Weakness 1)*
>
> To evaluate the quality of the automatically generated annotations, we conducted a **manual verification process**. Specifically, a team of **20 annotators** reviewed a total of **404,112 samples**, identifying **29,457 erroneous annotations**, corresponding to an overall error rate of **7.29%**, which we consider acceptable for large-scale automatic data construction. Due to space limitations, the detailed error distribution table is provided in the anonymous link:
> https://anonymous.4open.science/r/icml_rebuttal-39D0/Error_table.md
>
>
> ### *Zero-shot Performance of Qwen3-VL-235B (Weakness 1)*
>
> We report the zero-shot performance of **Qwen3-VL-235B** on multiple visual reasoning benchmarks:
> https://anonymous.4open.science/r/icml_rebuttal-39D0/Qwen3vl-235B_results.md
>
> ### Response to *Synthetic Bias in Benchmark Construction (Weakness 2)*
>
>  To improve the quality of **SV-QA**, we did not directly use the automatically generated outputs as final annotations. Instead, the benchmark data were further **manually inspected and corrected** to reduce annotation errors and improve question quality.Some qualitative examples of benchmark can be found in **Appendix D, Figures 9--14**. We will clarify this more explicitly in the revision.
>
>
> ### Response to *Concerns on Semantic Attributes (Weakness 2 and Key Question 2)*
>
> To address the concerns, we additionally evaluate SLVR on **VisualPuzzles** and three **math benchmarks**.
>
> Due to space limitations, the detailed results are provided at:
> https://anonymous.4open.science/r/icml_rebuttal-39D0/VisualPuzzles_math.md
>
>
> ### Response to *Limited Performance Gain over LVR  (Weakness 3)*
>
>
> We thank the reviewer for the concern.
>
> On standard benchmarks (Table 1), SLVR achieves clear improvements such as **+11.2 (OKVQA)**, **+24.7 (VizWiz)**, and **+12.8 (ChartQA)**, while remaining competitive on others.
>
> More importantly, beyond Tables 1 and 5, we further evaluate on **reasoning-intensive benchmarks**:
> - **Additional benchmarks:** strong gains on complex tasks (e.g., **+11.6 MMMU, +35.1 MMBench**)
> - **VisualPuzzles:** consistent improvements across all categories (**+4.8 overall**)
> - **Math benchmarks:** steady gains (e.g., **+6.2 MathVerse, +5.9 MathVision**)
>
> Full results at: https://anonymous.4open.science/r/icml_rebuttal-39D0/lvr_vs_slvr.md
>
> ### Response to *Key Question 3*
>
> We attribute the performance drop mainly to ambiguous questions or cases requiring external knowledge. We will focus on improving these aspects in future work. Example failure cases are provided at:
> https://anonymous.4open.science/r/icml_rebuttal-39D0/failure_cases.md

---

> > ### Author Rebuttal · Reviewer_AY3d · 2026-04-03
> >
> > Thank you for the rebuttal. I really appreciate the added experiments and the manual verification result, but my main concerns remain.
> >
> > Regarding the annotation quality, the reported **7.29% error rate** is useful, but more details should be provided. The author said "the benchmark data were further manually inspected and corrected to reduce annotation errors and improve question quality." Does it mean that all errors are corrected? The original paper didn't mention anything about this process. **29,457 erroneous annotations** is not a small number, and the rebuttal does not provide enough detail on how the manual verification and correction were conducted (e.g. procedure, time, cost).
> >
> > The rebuttal also does not resolve my concern about the semantic design. The paper uses a fixed templated attribute schema and collapses the full attribute set into a single 4096-dimensional embedding. The rebuttal provides additional benchmark results, but does not answer the core weakness of whether this template-based method is too rigid and lossy for rich semantics and complex reasoning settings. This concern is also reflected in the controlled experiments, where the gains on HRBench are very small, if any.

---

> > > ### Author Response · Authors · 2026-04-04
> > >
> > > We sincerely thank the reviewer for the detailed feedback and for carefully examining our rebuttal.
> > > ### Response to Annotation Quality
> > >
> > > Clarification.
> > > The statement “manually inspected and corrected” refers to the SV-QA benchmark, not SLV-Set.
> > > SV-QA had already been manually corrected beforehand. Since it is a small dataset (391 samples), the workload was manageable.
> > >
> > > In SV-QA, we identified 27/391 problematic cases:
> > > - 19 cases where the generated question had overly high overlap with the original question
> > > - 8 cases with hallucinated questions
> > >
> > > We will clarify this process more explicitly in the revised paper.
> > >
> > >
> > > SLV-Set verification.
> > > In contrast, SLV-Set was not manually corrected in advance. Its verification was conducted during the rebuttal period as a post-hoc quality assessment.
> > >
> > > - Procedure: annotators examined the image, region profile, and the two associated questions, and verified their consistency (grounding, semantic attributes, and QA alignment). If an issue was found, the sample was recorded and categorized.
> > > - Time: about 4 days (Mar 25–29)
> > > - Cost: about $3,000 for annotators
> > >
> > > We used a 20-person team, with each annotator reviewing roughly 20k samples. This process identified 29,457 erroneous annotations, corresponding to an error rate of 7.29%. We believe this level of noise is acceptable for large-scale automatically constructed training data. At the training stage, some degree of noise is unavoidable, and the consistent performance improvements across multiple benchmarks suggest that this level of noise does not significantly affect model learning.
> > >
> > > ### Response to Concern on Whether the Template-Based Design Is Too Rigid and Lossy
> > >
> > > We thank the reviewer for this important concern. We address this concern from two perspectives.
> > >
> > > **(1) Why the design is not inherently rigid or lossy.**
> > > Our semantic design should be understood together with the two-stage training strategy.
> > >
> > > A useful analogy is standard Chain-of-Thought training: in many recent MLLMs, the SFT stage first teaches the model to enter a reasoning format such as `<think> ... </think>`, while the RL stage further optimizes this reasoning process. Our design follows the same logic. In Stage 1, the supervision is used to teach the model to enter the latent reasoning mode within `<sem_start>` and `<sem_end>`. Since latent-space SFT lacks explicit supervision signals, we introduce structured attribute descriptions as supervision. These attributes are relatively deterministic and grounded, which helps activate fine-grained perception and build structured region-level understanding.This also addresses the concern of being lossy. Prior work (e.g., *Think Silently, Think Fast: Dynamic Latent Compression of LLM Reasoning Chains*, NeurIPS 2025) has shown that latent representations can serve as more information-dense representations.
> > > In Stage 2, due to the nature of RL and our reward design, the latent representation is optimized toward what is most useful for downstream reasoning. Answer correctness encourages retaining semantically useful information, while multi-query consistency forces one shared latent to support multiple questions over the same region. Therefore, the final latent is refined toward richer semantics and more complex reasoning behavior.
> > >
> > > **(2) Experiment on Template Restrictiveness.**
> > > We construct a controlled comparison that isolates the effect of Stage 1. In the “LVR + Multi-Q” setting (Table 3), the model is directly optimized with GRPO on the same 2q data, without learning the semantic template in Stage 1.
> > >
> > > #### Standard benchmarks
> > >
> > > | Model | OKVQA | GQA | VizWiz | ChartQA | TextVQA | AI2D |Avg|
> > > |------|------:|----:|------:|-------:|--------:|----:|-----:|
> > > | LVR + Multi-Q | 49.9 | 54.5 | 57.4 | 69.0 | 72.3 | **76.2** | 63.2 |
> > > | SLVR-7B | **61.8** | **55.6** | **57.8** | **77.2** | **79.3** | 76.0 | **68.0**|
> > >
> > > #### SV-QA
> > >
> > > | Model | V* Q1 | V* Q2 | V* Both | HRBench-4K Q1 | HRBench-4K Q2 | HRBench-4K Both | HRBench-8K Q1 | HRBench-8K Q2 | HRBench-8K Both |
> > > |------|------:|------:|--------:|--------------:|--------------:|----------------:|--------------:|--------------:|----------------:|
> > > | LVR + Multi-Q | 80.6 | 77.5 | 65.4 | 68.4 | 81.4 | 58.6 | 60.4 | 71.4 | 49.7 |
> > > | SLVR-7B | **82.2** | **80.1** | **69.1** | **70.4** | **82.8** | **61.1** | **62.5** | **73.6** | **50.6** |
> > >
> > > Due to space limitations, we provide additional results on more complex reasoning:
> > > https://anonymous.4open.science/r/icml_rebuttal-39D0/visualpuzzles_math2.md
> > >
> > >
> > > Regarding HRBench, the relatively small gains on some single-query metrics (e.g., Q1) reflect a trade-off introduced by multi-query optimization: improving shared latent consistency across queries may slightly affect per-query optimality. Since our method enforces a unified latent representation, the more informative metric is Both. Notably, the improvement on Both is larger than that on Q2, indicating that the model more importantly improves joint correctness across querie.

---

### Decision · Program_Chairs · 2026-04-30

**Decision:**

Accept (regular)

**Comment:**

The paper proposes SLVR, a two-stage framework for multimodal latent visual reasoning that enriches region-level latent representations with attribute-level semantic information and aligns them across multiple queries grounded in the same visual region. It also introduces SLV-Set, a large-scale training resource with region-level semantic annotations and paired question-answer data, together with SV-QA, a benchmark for evaluating reasoning consistency under semantic variation.

The main concerns were the reliance on synthetic supervision, fairness of comparisons, the role of stronger teacher-generated annotations, and the flexibility of the semantic design. In response, the authors provided substantial additional experiments and clarifications, including fairer comparisons, broader benchmark results, scaling evidence, annotation-quality analysis, and targeted ablations. These additions resolved the concerns of multiple reviewers. Overall, the reviewers found the paper clearly written, technically well motivated, and meaningful in both its methodological and resource contributions.

The paper still has some issues, particularly its reliance on model-generated supervision and some open questions about the flexibility of the template-based semantic design. However, these concerns do not outweigh its strengths. For the final version, the authors should incorporate the additional rebuttal experiments, clarify the annotation and correction process for SLV-Set and SV-QA, and temper claims where the evidence remains limited.